# Polymer Nanoparticles and Nanomotors Modified by DNA/RNA Aptamers and Antibodies in Targeted Therapy of Cancer

**DOI:** 10.3390/polym13030341

**Published:** 2021-01-21

**Authors:** Veronika Subjakova, Veronika Oravczova, Tibor Hianik

**Affiliations:** Department of Nuclear Physics and Biophysics, Faculty of Mathematics, Physics and Informatics, Comenius University, Mlynska dolina F1, 842 48 Bratislava, Slovakia; veronika.subjakova@fmph.uniba.sk (V.S.); veronika.oravczova@fmph.uniba.sk (V.O.)

**Keywords:** polymer nanoparticle, nanomotors, antibodies, nucleic acid aptamers, targeted drug delivery

## Abstract

Polymer nanoparticles and nano/micromotors are novel nanostructures that are of increased interest especially in the diagnosis and therapy of cancer. These structures are modified by antibodies or nucleic acid aptamers and can recognize the cancer markers at the membrane of the cancer cells or in the intracellular side. They can serve as a cargo for targeted transport of drugs or nucleic acids in chemo- immuno- or gene therapy. The various mechanisms, such as enzyme, ultrasound, magnetic, electrical, or light, served as a driving force for nano/micromotors, allowing their transport into the cells. This review is focused on the recent achievements in the development of polymer nanoparticles and nano/micromotors modified by antibodies and nucleic acid aptamers. The methods of preparation of polymer nanoparticles, their structure and properties are provided together with those for synthesis and the application of nano/micromotors. The various mechanisms of the driving of nano/micromotors such as chemical, light, ultrasound, electric and magnetic fields are explained. The targeting drug delivery is based on the modification of nanostructures by receptors such as nucleic acid aptamers and antibodies. Special focus is therefore on the method of selection aptamers for recognition cancer markers as well as on the comparison of the properties of nucleic acid aptamers and antibodies. The methods of immobilization of aptamers at the nanoparticles and nano/micromotors are provided. Examples of applications of polymer nanoparticles and nano/micromotors in targeted delivery and in controlled drug release are presented. The future perspectives of biomimetic nanostructures in personalized nanomedicine are also discussed.

## 1. Introduction

The progress in current nanomedicine relates to extensive research focused on the development of polymer nanoparticles and nano/micromotors modified by antibodies or nucleic acid aptamers that can recognize cancer markers at the surface of the cells and can serve as a cargo for the transport of antisense nucleic acids inside the cell for the purpose of gene therapy [1]. Cancer is among the most serious disease causes of death worldwide. In 2018, there were 18.1 million new cases and 9.5 million cancer-related deaths worldwide. By 2040, the number of new cancer cases per year is expected to rise to 29.5 million and the number of cancer-related deaths to 16.4 million [2]. The survival rate of the patients substantially increases with early diagnosis of this disease. It should be also noted that when cancer is detected, a certain number of patients are asymptomatic. For example, in the case of lung cancer, up to 10% of patients belong to this group [3]. Currently, traditional techniques such as laboratory tests of human fluids (blood, urine) are mostly used for identification of specific cancer markers, for example the prostate specific antigen (PSA) [4]. These tests are performed in specialized laboratories. Imaging tests are rather effective tools for cancer diagnosis and include computer tomography (CT), magnetic resonance imaging (MRI), nuclear scans, bone scans, X-rays scans and ultrasonic scans. CT is a well-established method that uses an X-ray machine linked to a computer in order to receive scans at various angles for obtaining a 3D image of the certain organ. For better resolution, the contrast material is applied intravenously or orally [3]. MRI uses powerful magnets and radio waves to obtain several pictures in slices to form a 3D picture of the organ. In some cases, contrast agents are also used [5]. Nuclear scan is based on the application of a small amount of radioactive material that is injected intravenously and accumulates in certain tumors containing organs or bone. This scan is also known as a radionuclide scan. After a certain time, the radioactive material loses its radioactivity or is removed from the body through urine or stool [6]. The bone scan is a type of radionuclide scan and monitors abnormal areas or damage in bone [7]. Among imaging techniques positron emission tomography (PET) is also used, which allows for a detailed 3D picture of areas inside the body where glucose is taken up. This is because cancer cells often take up more glucose than healthy cells [8]. Ultrasound technique is a non-invasive method of diagnosis. The intensive ultrasound wave is used for obtaining sonograms of certain organs or tissues [9]. X-ray diagnosis is among the oldest traditional methods allowing for tumor identification. It is based on the different densities of tumor and healthy tissues [10]. Colonoscopy is a rather useful diagnosis method for the identification of colorectal cancer as well as for its non-invasive treatment [11]. Biopsy belongs to the most occurring tests. A small part of tissue is removed from the body and investigated by pathologists. This assay allows selection of further cancer therapy [12]. Liquid biopsy is another method of diagnosis applied especially for leukemia and gastric cancer [13,14]. Advanced methods of cancer diagnosis include detection of circulating cancer cells (CRC) as well as next generation sequencing (NGS) that allows for genetic characterization of malignancies [13]. The novel approaches in biosensor technology for early diagnosis of cancer is based on various methods of detection cancer cells with sensitivity of 10 to 100 cells/mL (see [15,16] for recent review). Novel trends are focused on the application of nano/micromotors driven by various physical mechanisms, e.g., by ultrasound, to move these structures inside the cells. For example, nanomotors consisting of gold nanorods and covered by graphene oxide (GO) were used as support for the immobilization of nucleic acid aptamers that selectively bind to certain cancer markers in the membrane of the cancer cell. The aptamer is modified at one end by a fluorescence probe that is quenched by GO. Interaction of aptamer with cancer markers caused its folding, moving the fluorescent probe away from the GO. This is accompanied by increase of fluorescence [17].

The treatment of the cancer uses various methods such as surgery, radiotherapy [18], immunotherapy, or chemotherapy [19]. In the case of immuno- or chemotherapy it is crucial to develop a method of targeted delivery of specific antibodies, antisense nucleic acids, or chemotherapeutic drugs inside the affected tissues [20]. This approach can help in avoiding the undesirable toxic effect of drugs on normal, healthy cells. For targeted therapy, various methods are currently being investigated. This includes metal or polymer nanoparticles, polymer micelles [21] and nano/micromotors modified by nucleic acid aptamers or antibodies [22] that recognize the cancer markers at the membrane of the cells. By endocytosis, the nanocarriers that load the drug are then moved inside the cell where the drug is released. Novel trends consist also in stimuli responsive drug release [23].

Current nanomedicine is focused on the application of various nanostructures for diagnosis and targeted therapy (the theranostics) of the cancer. For example, various types of inorganic and polymer nanoparticles as well as nanocomposites can help in the diagnostics of cancer using various imaging techniques and provide an alternative to common cancer chemotherapy [24]. In the case of metastasis of the cancer, the chemotherapy and immunotherapy are the only possible methods of cancer treatment. However, classical injection of chemotherapeutic drugs such as for example doxorubicin affects not only the cancer cells, but also other healthy cells and tissues. Therefore, the cargo that delivers the chemotherapeutic drug should selectively recognize the cancer cells. It is well established that various cancer markers are over expressed at the cancer cell membranes, for example protein tyrosine kinase 7 (PTK7) in HeLa cells isolated from cervical cancer [25]. Thus, modification of the nanostructures such as nanoparticles or nano/micromotors by receptors that can recognize these cancer cells, is of high advantage for targeted drug delivery. Among possible receptors the nucleic acid aptamers and antibodies can be used for modification of nanoparticles or nano/micromotors. The nanocarrier can interact selectively with the cancer marker at the surface of the cell and transfer the drug inside the cell by endocytosis, as schematically shown in Figure 1.

The above-mentioned active transport of nanoparticles is more effective in comparison with the passive mechanism. The latter is based on the enhanced permeability and retention (EPR) effect due to leakage of vasculature surrounding the tumors [21]. In active cancer therapy, among possible cargo, various nanostructures can be used, such as inorganic nanoparticles (NPs), polymer NPs, biomimetic NPs and nano/micromotors. Among the receptors that recognize the cancer markers, the DNA/RNA aptamers are of high advantages. The aptamers are single stranded DNA or RNA that in a solution fold into a 3D structure forming a binding site for the respective cancer marker. The method called Cell SELEX (Systematic Evolution of Ligands by Exponential Enrichment) is well established for the selection aptamers that recognize cancer markers at the membranes of the cancer cells. The nucleic acid aptamers provide a big advantage in comparison with antibodies due to their no immunogenicity, higher stability, and easy modification [1].

This review presents the current state of the art in the development of polymer nanoparticles and nano/micromotors for targeted drug delivery and controlled drug release. Special focus is on the application of nucleic acid aptamers—the unique molecules that can recognize the cancer markers with high specificity. We also discuss the mechanisms of stimuli-responsive drug release, toxicity of nanomaterials, their clinical applications as well as future perspectives. This review represents the most recent state of the art in the targeted drug delivery and for the first time provide a complex inside on the application of polymer nanoparticles and nano/micromotors in the targeted drug delivery. The targeted drug delivery is still in a premature stage. The mechanisms of interaction of nanoparticles and nano/micromotors with the cell membranes are not yet known in sufficient details. In addition, the clinical trials focused on the application of nanomaterials are in a very early stage. It is expected that the review can be useful not only for researchers but also for clinicians who are interested in the application of nanotechnology in the theranostics of cancer.

## 2. Polymer Nanoparticles

### Preparation, Structure, and Properties of Polymer Nanoparticles

Polymers are compounds with unique properties making them suitable candidates for drug loading and delivery. For almost three decades, polymeric materials have been studied in targeted drug delivery, bringing important results and developments. Polymers suitable for nanoparticle preparation can be of synthetic or biological origin. In drug delivery, an important issue is biodegradability of the material used for nanoparticle preparation. In general, biodegradable materials have an immense advantage and are preferred in the outmost applications in biomedicine. Overall, it is important, that after the degradation of polymers, the remaining monomers should be non-toxic. Amongst usually used biocompatible, resorbable, non-toxic monomers in the preparation of micro- or nanoparticles are poly(d,l lactic acid) (PLA) and poly(d,l lactide-*co*-glycolide) (PLGA), which are metabolised trough the Krebs cycle [26].

Polymer nanoparticles (PNPs) can be of various morphology and composition in the core or periphery, for example dendrimers, polymeric micelles, polymer-lipid hybrid and polymeric nanoparticles (Figure 2). Depending on the method of synthesis, a drug can be either physically entrapped in the core of the PNPs or covalently bound to the matrix of PNPs.

Dendrimers represent an attractive hyperbranched 3D object for biomedicine. Their greatest advantage is the ability to have multiple functional groups on their surface. Thanks to their structure and high degree of branching they can encapsulate cargo within central cavities based on various chemical conjugation methods, making them supramolecular complexes. Their 3D spherical shape, nanometer size, lipophilicity, monodispersity and ability to penetrate cell walls make them an ideal delivery system. As a delivery system, often encountered are diaminobutyric polypropyleni-mine (DAB), poly (amido-amine-organosilicon) (PAMAMOS), poly (Lysine), polyamidoamine (PAMAM), and poly (propylene imine) (PPI) [27].

Polymeric micelles are formed by amphiphilic block copolymers and contain a hydrophobic core and a hydrophilic corona. Owing to their structure, it is possible to chemically conjugate or entrap various biomolecules including proteins, nucleic acids, peptides, and phospholipids. Frequently used block polymers include hydrophobic, biocompatible, and biodegradable segment polylactide (PLA) or hydrophilic, water-soluble poly (ethylene glycol) (PEG), 1-vynil 2-pyrrolidone (VP), polyacrylic acid (PAA) or poly *N*-izopropylacrylamide (NIPAAM). By functionalization of the surface of the micelles and with the unique possibility to transport hydrophobic bioactive therapeutics, polymeric micelles represent an effective solution for targeted drug delivery [28]. Luo et al. [29] presented a study concerning dual pH/redox responsive polymeric micelles aimed for targeted drug delivery and controlled release. For the micelles, two kinds of amphiphilic block copolymers were used: poly(ethylene glycol) methyl ether-grafteddisulfide-poly(β-amino esters) (PAE-ss-mPEG) and (poly(ethylene glycol)methyl ether-b-poly(β-amino esters) (mPEG-b-PAE). After synthesis of polymeric micelles, these nanocarriers were later loaded with doxorubicin with exceptional drug-loading efficacy. The study revealed that the micelles successfully released the therapeutic drug after entering cancer cells, where at low pH and high glutathione concentrations the micelles swell and disassemble, releasing the load. The in vitro studies revealed high cytotoxicity of the nanoparticle complex against HepG2 cells, which proves them to be a potential targeted drug delivery complex [29]. Recently, casein nanoparticles have also been reported as a novel system for drug delivery. The menthol-modified casein nanoparticles loaded by 10-hydroxycamptothecin more deeply penetrated into the brain glioma tumor in comparison with unmodified nanoparticles [30].

After years of experiments and studies in the field of nanotechnology, precise methods for synthesis have been presented. There are two types, top-down or bottom-up approaches. The “top-down” method represents the conversion of bulk material into nanoparticles. This includes photolithography, electron beam lithography, anodizing, ion and plasma etching, and techniques involving material crushing. Another approach to nanoparticle synthesis is the, bottom-up” method, which involves the coalescence or assembly of atoms and molecules that make up nanoparticles. This method is represented by self-assembling monomers of polymer molecules, chemical or electrochemical nanostructured precipitation, laser pyrolysis, chemical evaporation, plasma/flame spray synthesis, and biosynthesis [31].

In general, methods of nanoparticle synthesis can be divided into three groups: physical, chemical, and biological. Recently, several techniques for the synthesis of polymeric nanoparticles have been developed. They are classified depending on whether the formation process involves a polymerization reaction or whether polymer nanoparticles (PNPs) are formed directly from the macromolecule (Figure 3). The possibilities of PNPs synthesis can be divided into two main groups:(1)Preparation of polymer nanoparticles from formed polymers(2)Production of polymer nanoparticles by polymerization of monomers

The choice of the method for the preparation of nanoparticles is made on the basis of several factors, such as the desired size, the type of polymer system, the field of application, etc. [32].

Preparation of nanoparticles from polymers can be achieved by the following methods. Solvent evaporation is known to be the most common technique of PNPs synthesis. Two main techniques can be adapted whilst performing solvent evaporation: a) preparation of single-emulsions (e.g., oil-in-water) b) double emulsions (e.g., (water-in-oil)-in-water). These methods are based on ultrasonication or high-speed homogenization, followed by evaporation of the solvent by continuous stirring at room temperature or under reduced pressure. The additives, such as surfactants, can be removed by ultracentrifugation and washing with distilled water. After removal of the solvents and additives, the final product is lyophilized [32]. Kizilbey [33] in 2019 presented a rutin-loaded PLGA nanoparticle prepared by single-emulsion solvent evaporation. Rutin is a bioactive molecule, which has wide applications in pharmacological or food products. Since this molecule has poor water solubility and low bioavailability, it was dissolved in propylene glycol, and further entrapment into poly(d,l-lactide-*co*-glycolide) nanoparticles was performed by oil-in-water single-emulsion solvent evaporation. Results showed that their design of encapsulation is efficient and could be further applied in targeted delivery of poorly water-soluble drugs.

Nanoprecipitation is a method developed by Fessi et al. [34] in 1989 for the synthesis of PNPs. This method is based on interfacial deposition of a polymer following the displacement of a semipolar solvent, which is miscible with water, from a lipophilic solution. The decrease of interfacial tension between the two phases, caused by rapid diffusion of solvent into non-solvent phase, results in the increase of surface area and leads to the formation of nanoparticles. As Liu et al. reported in 2020 [35], simple and robust sequential nanoprecipitation is a suitable method for enhancing drug loading and produces a stable drug-core polymer shell PNPs with high amount of loaded drug (up to 58.5%). Although to achieve such a loading, organic solvents had to be used, which are hazardous to the environment and to biosystems [32]. Several other recent studies reported synthesis of polymeric nanoparticles using nanoprecipitation method. For example, for biomedical purposes, electroactive poly(vinylidene fluoride-*co*-trifluoroethylene) microspheres were synthesized by Macedo et al. [36]. In the article of Jiang et al. [37], amphiphilic random and block copolymers synthesized by nanoprecipitation were compared. It has been shown that the composition and hydrophilic-lipophilic balance within the polymeric particles greatly affected the loading capacity. The application of disulfide moieties within the polymeric particles allowed controlled cargo release. Further information about nanoprecipitation methods used for the synthesis of polymer nanoparticles for drug delivery can be found in a recent review by Liu et al. [38].

Salting out method is a modified version of the emulsification/solvent-diffusion approach; it does not require high-energy and high-pressure input in comparison with methods using toxic solvents. Therefore, it is a greener technique for the synthesis of PNPs [39]. This method uses a modified version of the emulsion process conjoined with a salting-out process avoiding chlorinated solvents and surfactants. The emulsion is created with a polymer solvent, which is usually miscible with water, and emulsification of the polymer solution is achieved in the aqueous phase without the engagement of high-shear forces, similar to an Ouzo effect [40], by dissolving high concentrations of either salt or sucrose for salting-out effect in the aqueous phase. As these chemicals are dissolved in water, the miscibility properties of the water with other solvents are modified, leading to the precipitation of the polymer dissolved in the droplets of the emulsion [32]. The addition of water to the emulsion under stirring leads to the migration of the water-soluble solvent from the emulsion droplets, assisting in nanoparticle formation [32,41]. Finally, purification of nanoparticles by centrifugation or cross flow filtration is applied [41]. Qu et al. [42] synthesized cabazitaxel-loaded human serum albumin nanoparticles (Cbz-NPs) for the treatment of prostate cancer by the salting-out method to overcome the use of chlorinated organic solvents. The method they proposed was successful in creating Cbz-NPs with narrow particle size distribution, proven drug loading (4.9%), and great blood biocompatibility. This study proved the suitability for future use in the clinical therapy of the proposed nanoparticle ensemble for its prolonged blood circulation and enhanced accumulation in tumor tissue of prostate cancer.

Dialysis represents a simple and effective tool for the synthesis of polymeric nanoparticles. Dialysis often requires the use of organic solvents [32]. The polymer is dissolved in a water miscible organic solvent and placed inside a dialysis tube with the desired molecular weight cut-off. The organic phase diffuses trough the dialysis tube into the aqueous phase, decreasing the interfacial tension among them. Subsequently, the formation of homogenous suspension of nanoparticles is observed after the displacement of the solvent inside the membrane, following progressive aggregation of polymer owing to loss of solubility [32]. The dialysis method was used in the study of Shakeri et al. [43], where prepared carvacrol loaded polyhydroxybutyrate (PHB) nanoparticles were spherical, 140 nm in diameter with monomodal distribution, and with an entrapment efficacy of 11%. On the other hand, supercritical fluid technology (SCF) offers a greener method of synthesis of polymeric nanoparticles. The SCF can be used either as (A) solvent to dissolve the drugs, polymer, or other components; (B) antisolvent for the induction of precipitation of polymer particle components; (C) processing additive, which can contribute to high solute or solvent mobility, higher saturation level or melting point depression. There have been reported many methods where SCF contributes to production techniques, such as rapid expansion from supercritical solution (RESS), supercritical antisolvent (SAS) and as well as particles from gas-saturated solutions (PGSS) [44]. The RESS technique altogether with rapid expansion of a supercritical solution into a liquid solvent (RESOLV) are based on the same principle. The polymer is dissolved in a supercritical fluid, which is then subjected to rapid expansion through a nozzle in ambient air. Subsequently, the sudden reduction in pressure leads to a supersaturation promoting homogenous nucleation and the formation of nanoparticles. The expansion in the RESOLV technique occurs in a liquid solvent instead of air, which hinders particle growth, thus leading to the synthesis of nanoparticles. The disadvantage of these methods is low solubility of polymers in supercritical fluid, as well as the difficulty to control particle size [45]. However, SCF methods have been presented in various biomedical research, such as PGSS method that has been used in the study of Tokunaga et al. [46]. In this work, microencapsulation of drugs with enteric polymer Eudragit L100 for controlled release by changing pH value has been reported. Another research concerning SAS coprecipitation has been conducted by Montes et al. [47]. In this work, coprecipitation of ibuprofen with the polymers poly(l-lactic acid) and Eudragit L100 by SAS method was presented, leading to slower and more controlled release in comparison with unprocessed ibuprofen. Detailed information about SCF methods can be found in a recent review of Chakravarty et al. [44]. Preparation of polymer nanoparticles from monomers can be achieved by using the following methods of polymerization. 

The emulsion method represents the simplest synthesis procedure for polymeric nanoparticles. It is the most common technique for the synthesis of polymeric nanoparticles with a size less than 500 nm [48]. Plenty of pharmacological studies based on polymeric nanoparticles prepared by emulsion methods have already been conducted. For detailed information about this topic, please see the recent review by Jenjob et al. [49]. The single-emulsion method is based on dissolving the polymer in an organic solvent immiscible in water. Under subsequent stirring, after applying stabilizer, an emulsion in the aqueous phase is formed. To remove the traces of either solvent, free drug or stabilizing chemicals various cleaning methods are advised before lyophilization. In most cases, it is necessary to optimize the protocol several times to acquire the desired output [45]. The nanoparticle mean size and thickness of the supersaturated region can be affected by preparative variables, such as polymer concentration or stirring rate [50]. Single-emulsion method creates nanoparticles with low encapsulation efficiency of hydrophilic drugs. Double-emulsion methods do not dispose with this issue, although for the encapsulation of hydrophilic drug, the use of organic solvents is often necessary [45]. In the study of Liu et al. [51], modified double emulsion method was applied for the preparation of regular spherical PLGA nanoparticles with diameters of 200 to 300 nm. These nanoparticles showed exceptionally high encapsulation efficiency (>80%) and loading (6.5% *w*/*w*) of hydrophilic drug daunorubicin. 

Interfacial polymerization is a known well-established method for the synthesis of polymer nanoparticles, which involves polymerization of two highly reactive monomers, which are dissolved in two immiscible phases. The reaction occurs at the interface of the two solutions [52]. This method is widely used in various fields, such as industrial preparation of conducting polymers or in pharmaceutical industry for the preparation of encapsulated products [32]. Another example of the use of interfacial polymerization is the preparation of thin-film nanocomposite with integrated zwitterionic polymeric nanoparticles [53]. The nanofiltration membrane was permeable for pure water but demonstrated high metal ion removal for Pb^2+^ and Cd^2+^ ions. Bossion et al. [54] prepared non-isocyanate polyurethane soft nanoparticles by overcoming the encumbrance of miniemulsion polymerization technique, which required non-trivial control of the polymerization conditions due to the incompatibility of monomers containing isocyanate and water. For the synthesis of nanoparticles, interfacial polymerization was applied using non-isocyanate route, which minimized the side reactions with water. The polyurethane nanoparticles size was between 200 and 300 nm. This polymerisation method allows for the immobilization of functional groups on the nanoparticles as well, making this synthesis method suitable for creating particles destined for drug delivery.

Controlled/living radical polymerization (C/LR) or Reversible-deactivation radical polymerization (RDRP): is basically a chain polymerization, where the active polymer chain end is a free radical. It is a chain reaction propagated by radicals that are deactivat-ed reversibly. The synthesis process has three stages: initiation, propagation and termination. Developments in polymer synthesis techniques, in particularly ionic and radical polymerization methods, allow for the preparation of block copolymers with a well-defined composition, molecular weight, and complex architecture. The growing range of available block copolymer structures includes linear block copolymers, graft copolymers, dendritic, star, cyclic polymers, etc. [55]. Block copolymers have widespread use for their ability to encapsulate various pharmacologically active agents, which is beneficial for targeted drug therapy. Depending on the application and the required release profile of the active chemical, stable delivery systems are needed. Amphiphilic block copolymers are particularly stable when diluted and exhibit prolonged circulation in the bloodstream [56]. C/LR polymerization can be divided into several subtypes. For example, one of the subtypes is degenerative transform system (RAFT) polymerisation. Pourjavadi et al. [57] presented pH and thermal dual-responsive poly(NIPAM-*co*-GMA)-coated magnetic nanoparticles synthesized trough surface initiated RAFT polymerisation aimed for targeted drug delivery. For further information about C/LR, please see the review by Zetterlund et al. [58].

Many studies have reported the advantage of poly (ethylene glycol) (PEG) as the outer nanoparticle shell that protects the core by steric stabilization [59]. Steric stabilization is a mechanism that explains the ability of certain additives to inhibit coagulation of suspensions. These additives usually represent different types of hydrophilic polymers and surfactants with hydrophilic chains. The additives cover the system in such a way that the long loops and ends protrude into the solution. In this way, the system is sterically stable even at elevated salt concentrations or when the zeta potential is almost zero. PEG-ylation is known to reduce the rapid clearance (amount of plasma that is purified from a given substance per unit time) of nanoparticles and maintain their viability in the bloodstream, thereby giving sufficient time to therapeutic agents to reach the target site [59]. The clearance of nanoparticles from the blood depends mainly on their size, shape, and rate of biodegradation [60].

## 3. Nanomotors

Over the last two-decades, nano/micromotors have been welcomed by a growing interest due to their promising application in biosensing, drug delivery, and therapy. Nano/micromotors are machines capable of converting energy from different sources to movement. Nano/micromotors can be classified based on the nature of machines into biological and artificial motors. Biological motors use chemical energy stored in adenosine triphosphate (ATP) that can be converted into mechanical work. Some examples of molecular protein motors using ATP energy are kinesin, myosin, dynein, bacterial flagella etc. [61]. Biological motors and microorganisms inspired the development of functional synthetic motors and also as a potential engine replacement [62]. In this review, we focus mainly on artificial motors. Depending on propulsion strategy, nano/micromotors can be driven by chemical fuel or by an external source (magnetic, ultrasound, light, electrical field), or by a combination of driven forces or biomolecular systems called hybrids. Many challenges and tasks of controlled motion in solutions and in biological fluids are caused by the Brownian motion and by dominancy of viscous forces leading into low Reynolds number [63]. 

Enormous attention and effort in the progress of nano/micromotors fabrication and application proves the numbers of review articles regarding many aspects of their potential biomedical applications [64,65] and their biocompatibility [66], ranging from drug delivery [67,68], intracellular sensing and delivery [69], to diagnosis [70] and therapy [71,72].

### 3.1. Preparation of Nanomotors

The selection of material for fabrication and designing of nano/micromotors can effectively enhance their functional ability and versatility [73]. Compressive overview of fabrication techniques in development of nano/micromotors to overcome their limitation in term of improving propulsion mechanism and effectivity was reported by Wang and Pumera [74].

For synthesis of nano/microstructures, a membrane template-assisted electrochemical deposition is commonly used. This is a cost-effective method for the formation of nanostructures with various dimensions and material composition. Generally, porous anodic aluminum oxide (AAO) and polycarbonate membranes are used for the preparation of nanorods, nanowires, and nanotubes [75]. Porous AAO membranes are formed by anodization in electrolyte solution, resulting in ordered hexagonal pores [76] Thus, a certain area of template membrane and pore density provides homogenous structures with the length controlled by the charge [74]. Electrochemical deposition set up including a reference and working electrode. Therefore, a thin conductive layer, usually gold, is sputtered on one side of the template membrane serving as a working electrode. To create a concave shape at the end of the nanostructure, a sacrificial layer is electrodeposited. For example, for synthesis of gold nanowires, a membrane with a pore size of 200 nm was used. Applied potential vs. Ag/AgCl reference electrode for deposition of Cu using cupric sulfate pentahydrate solution at the charge of 8 C was −0.9 V. Then, the Au was deposited using a gold plating solution at a charge of 4 C and potential of −1 V vs. Ag/AgCl reference electrode to gain nanowires with length of ~4 μm [77]. An important step of nanostructure fabrication is the removal of a sacrificial layer and template membrane. This is performed by mechanical polishing and chemical cleaning. For AAO membrane, sodium hydroxide is usually used, while for polycarbonate membrane methylene chloride is typically applied for chemical cleaning [74]. The scheme of preparation nanomotors by electrodeposition in porous membrane is presented on Figure 4.

Other methods were also used for synthesis of the nanostructures. The template-assisted electrodeposition allows for the synthesis of nano/microtube [78] as well as helical structures [79]. In order to provide desired functionality of nanomotors, other methods such as conventional physical vapor deposition and glance angle deposition are used to produce nanostructures of desired shape and composition [74].

One of the most common nanomotor systems are Janus particles, owing to their unique physical and chemical properties due to asymmetric structure [80]. Several methods such as physical vapor deposition, electrochemical deposition, and self-assembly were introduced and discussed for fabrication Janus particles with different shape, material, and size [80,81].

### 3.2. Propulsion Strategy

#### 3.2.1. Catalytic Propulsion

Autonomous self-propelled motors provide perspective in biomedical [82] and environmental applications [83]. Motion of catalytic nano/micromotors follows several propulsion mechanisms. The movement is based mainly on the decomposition of chemical fuel e.g., hydrogen peroxide to water and oxygen on asymmetric bimetallic nanorods due to proton gradient among their surfaces. Platinum (Pt) is widely used for catalytic reaction and the motion forward Pt end is caused by the generation of an electric field through electron flow (Figure 5). This process is called self-electrophoresis [84]. In 2004 and 2005, Mallouk’s [85] and Ozin’s [86] groups demonstrated that bimetallic nanowires Au-Pt and Au-Ni were propelled by electrocatalytic decomposition of H_2_O_2_ by Pt and Ni segments, respectively. The bubble propulsion is common for nanomotors with microtube shape. The inner side of the tube usually consists of Pt catalyst, the fuel e.g., H_2_O_2;_ enters inside to tube producing oxygen bubble (Figure 5B). Similarly, as for bubble propulsion as well as for diffusiophoresis, the direction of motion is away from catalyst site. Diffusiophoresis is another approach for propulsion based on asymmetric nanostructure resulting accumulation of reaction products on one side. Then, the movement is caused by asymmetric gradient of products (Figure 5C). The motion toward to catalyst can be explained by interfacial tension, where products are accumulated on the Pt side, creating interfacial tension gradient [84,87].

The motion of self-propelled motors is, however, disordered. Therefore, addition of e.g., a magnetic element, allows for controlled guidance by the magnetic field [88]. In order to avoid high toxicity of H_2_O_2_, alternative biocompatible fuels such as urea [89] or glucose [90] were used. Because of the limitation of autonomous motors for in vivo applications, the enzyme powered motors are preferred candidates for applications in the biological environments [91].

#### 3.2.2. Ultrasound Propulsion

Acoustic radiation has been shown as a very useful and essential tool in medicine. Since 1990s, it is widely used in diagnostic imaging and in therapeutic applications due to high biocompatibility. [92]. The development of acoustic-powered machines promises a potential toward a variety of biomedical applications thanks to a noninvasive and on demand controllable propulsion [93,94]. Ultrasound waves as driven force for particles were demonstrated by Mallouk’s group in 2012 [95]. Ultrasound propulsion was generated by a ceramic transducer with a frequency of 3.7 MHz. The behavior of polymeric and metallic spheres and microrods was described under acoustic field. Metallic rods achieved a speed of ~200 μm/s in axial directional motion, which is explained through a self-acoustophoresis mechanism. Different types of nano/micromotor ultrasound-driven motion are shown in Figure 6A. Concave and convex curvature at the ends of nanorods led to this motion due to asymmetric scattering of acoustic waves. Activity of propelled nanomotors was not significantly affected by high ionic strength. Later, acoustic propulsion of nanomotors (~300 nm in diameter and ~3 μm long) was demonstrated in Hela cells as a model for a living system. Hela cells remained viable even after the application of ultrasound and internalization of nanorods. Motion with lower speed was observed inside the cells compared to extracellular side [96]. An alternative approach brought Kagan et al. [97] was based on the vaporization of biocompatible perfluorocarbon fuel by ultrasound. Perfluorocarbon emulsion was bound inside of metallic microbullets (length 40 μm, diameter 2.5 μm). Microbullets were capable of penetrating deeply through dense material of tissue. The average speed of the ultrasound triggered microbullets reached over 6 μm/s. Nadal and Lauga [98] presented a physical mechanism based on the theoretical calculation of the effect of acoustic field on asymmetrically shaped particles. The motion was caused by steady streaming, leading into the final drive speed along the axis of symmetry of the particle and perpendicular to the direction of oscillation. The dimensional velocity can be described by Equation (1):(1)$$v_{k\;}^{∥}=\epsilon R_{e\;}V^{⊥}v_{k}^{\left(1,1\right)}$$
where ϵ is the dimensionless small shape parameter, *R_e_* is the Reynolds number, $V^{⊥}$ is the amplitude of the oscillation of acoustic field, *ν_k_*^(1,1)^ with the superscript (1,1) is the leading-order dimensionless propulsion speed [98].

The effect of the shape and density of the bimetallic particles in acoustic propulsion was also studied. Bimetallic nanorods, e.g., Au-Ru, Au-Rh, moved toward the lower material density, while with similar densities of material, the motion was directed to the concave shape. The length of nanorods decreased the speed and the lighter single metal nanorods were propelled faster [99]. Numerical calculation and simulations of nano and microparticles with rounded or pointed and filled or hollow shape exposed to ultrasound waves were also investigated. Negligible effect of a cavity in particles was observed, whereas a pointed shape increased propulsion speed [100].

Many possibilities of biomedical applications of ultrasound-propelled nanomotors have been successfully demonstrated. To increase the drug loading capacity, porous gold nanowires were developed. Doxorubicin was loaded through electrostatic interaction with an anionic coating and drug release was controlled by near-infrared light due to photothermal effect. Porous nanomotor-propelled by ultrasound towards the Hela cells and releasing doxorubicin was monitored by fluorescence measurement [101]. Ultrasound propelled nanomotors have shown enhanced intracellular detection for miRNA-21 in MCF7 breast cancer cells [102], as well as an active and rapid effectively delivery of siRNA for gene silencing [77], Cas9/sgRNA complex for gene knockout [103], caspase-3 enzyme for apoptosis [104], oxygen into J774 macrophage cells [105] or metallodendrimers [106].
Figure 6Examples of different propulsion strategies of nanomotors powered by an external sources (**A**) Ultrasound, (**a**–**c**): schematic illustration of the motions of metal microrods in a 3.7 MHz acoustic field. Shown motion: axial, directional motion, in-plane rotation, chain assembly, axial spinning, and pattern formation, especially ring patterns (**d**,**e**): Dark field images of typical chain structures and ring patterns that were formed by Au and AuRu rods. Reproduced from [95] with permission of the American Chemical Society. (**B**) Magnetic: (**a**) Scheme of hybrid magneto-acoustic nanomotors and dual propulsion mode driven by magnetic and ultrasound fields. (**b**) SEM image of a magneto−acoustic hybrid nanomotor. Scale bar: 500 nm. Reproduced from [107] with permission of the American Chemical Society. (**C**) Light: Schematic illustration of the light-driven nanomotor. An n+-Si (green) shell was formed on a p-Si core (red) by thermal diffusion doping of phosphorous and platinum (yellow) nanoparticles were deposited on the surface as an electrocatalyst. Reproduced from [108] with permission of WILEY-VCH. (**D**) Electric field: (**a**) Scheme of 3-D orthogonal microelectrode setup **(b**) Speed of the nanomotor in the cargo delivery process. Reproduced from [109] with permission of American Chemical Society.
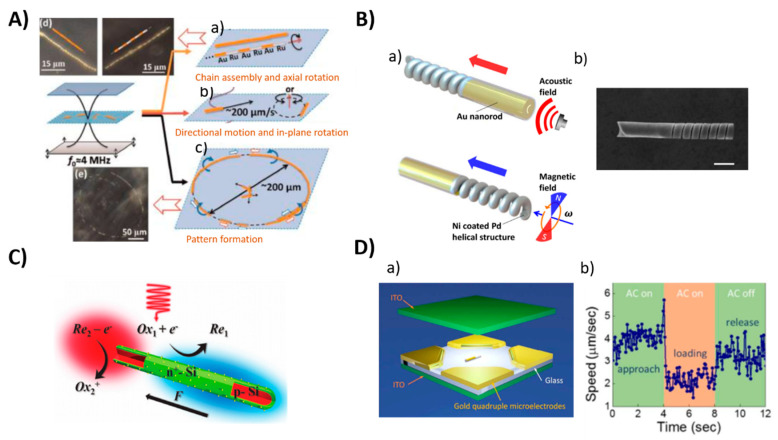



#### 3.2.3. Magnetic Propulsion

The design of magnetically driven motors is mostly inspired by prokaryotic and eukaryotic organisms having flagella, where rotational motion is transferred into translational motion. The magnetic field allows for a variety of swimming mechanisms for nano/micromotors depending on their shape (helical, flexible, surface walkers) [110]. In addition, the magnetic field for propulsion can be categorized as oscillating and rotational, resulting in a different mechanism motion [111]. Artificial flagella consisting of helical tail and magnetic head were controlled under low strength rotating magnetic field. At the low frequency of applied external field, a linear dependence of the translational velocity vs. magnetic field strength was observed. Soft-magnetic head affected magnetic torque and propulsive force of artificial flagella [112]. Furthermore, Gao et al. [113] demonstrated fabrication of nanowires with gold head and nickel tail joint with the flexible Ag due to partial dissolution in H_2_O_2_. Swimming devices deform their shape to actuate the nonreciprocal motion. Owing to Ag bridge, a mechanical deformation of nanowires under a rotating magnetic field was possible. The motion can be controlled by tuning the size of the nickel and gold parts and modulating the magnetic field. Flexible nanowires moved in solutions of high ionic strength as well as in biological urine sample with negligible velocity limitation. Later, the same group extended this study for biomedical applications and reported cargo-loaded magnetic nickel-silver nanoswimmers for targeted drug delivery. Directed delivery of drug loaded polymeric particles to Hela cells were performed through a microchannel. The effect of payload size was examined experimentally and compared to theoretical calculation [114]. In addition, magnetic field has been used as a guide for ultrasound propelled nanomotors with nickel segment offering movement along preselected paths [115]. Liu et al. [116] described a drug delivery strategy using magnetically powered wormlike mesoporous silica nanotubes decorating with CoFe_2_O_4_ magnetic particles. Manipulation of magnetic helical nanoprobes in living cells confirmed strong anisotropy and heterogeneity of the cellular interior. Despite the inhomogeneous environment in the cytoplasm, magnetic nanomotors could be controlled very precisely, offering drug delivery to specific location [117].

The Wang’s group developed hybrid ultrasound and magnetic propelled nanomotors (Figure 6B) [107]. The hybrid nanomotor is based on two segments—the Ni coated Pd magnetic nanospring and a concave Ni coated Au nanorod. The speed of the nanomotors was controlled by the voltage applied to the piezoelectric transducer (resonant frequency 2.66 MHz) or by frequency of rotating magnetic field, respectively, and was in the range of several µm/s. For example, the magnetic nanomotor moved in a direction toward Au end with speed from 7.6 to 15.9 µm upon changes of frequency from 100 to 200 Hz, respectively. The acoustic driven nanomotors moved in opposite direction with speed from 8.1 to 22.3 µm/s at an ultrasound voltage amplitude of 2 and 6 V, respectively. The advantage of the hybrid nanomotors was in the possibility of controlled direction of propulsion and operation in the bioliquids with high ionic strength, including blood.

The speed of nanomotors can be regulated by an external field. For magnetic helical structures, the speed depends on geometric parameters as well as the rotation frequency of the field, and can be described by Equation (2): (2)$$v=\frac{\left(\xi_{⊥}-\xi_{∥}\right)\sin\theta \cos\theta }{2(\xi_{⊥}\sin^{2}\theta +\xi_{∥}\cos^{2}\theta )}d\omega$$
where *ξ*_⊥_ and *ξ*_‖_ are the drag coefficients perpendicular and parallel to the helical axis, respectively, θ is the helix angle, *d* is diameter of the helix, and *ω* is the rotational frequency [107].

#### 3.2.4. Light and Electric Propulsion

The light is the other external stimuli for the propulsion of nanomotors with many advantages over controlling on demand with excellent spatial and temporal resolution, noninvasive treatment, and compatibility with biological samples. Light driven motors are made from photothermal, photocatalytic and photochromic materials, allowing photochemical reaction or thermal conversion due to the absorption of light energy [118,119,120]. In general, the velocity (*U*) of the nanomotor can be described in Equation (3):(3)U=b∇Y
where *b* is the velocity coefficient and ∇Y is gradient of asymmetric field around nanomotors that can include pressure, electric potential, solute concentration, or temperature [118]. 

Similarly, the electric field offers precise switching and efficient manipulation of nano/micromotors. Light and electric field is often used as an additional external source for the propulsion of nanomotors [119,120,121]. For example, Wang et al. [108] demonstrated light-powered silicon based nanomotors. The structure of nanomotors had a core-shell p-n junction with Pt nanoparticles as the catalyst on the surface, and as the redox couple 1,4-benzoquinone/hydroquinone and H_2_O_2_ were used (Figure 6C). Upon light exposure, a photovoltage is generated across p–n junction of silicon nanowire, which propels the electrochemical redox reaction and produces positively charged proton (H^+^) and negatively charged hydroxide (OH^−^) on the p-type core and n-type shell.

Guo et al. [109] reported the application of the electric field for manipulation of Pt-Au catalytic nanomotors to control the cargo capturing, delivery, and release to the special microdock as well as for the integration with nanoelectromechanical system (Figure 6D). The approach is based on the applied AC and DC electric field in three dimensions. Inverse linear dependence between the speed and size of nanomotors was observed and confirmed.

## 4. DNA/RNA Aptamers: Structure and Properties

In recent years, the development of new recognition elements for biosensor technology and targeted drug delivery is of increased importance. Among them, the discovery of nucleic acid aptamers (DNA or RNA) was a revolutionary step that promises high improvements in cancer diagnostics and therapy. Nucleic acid aptamers were discovered 30 years ago together with the method of their selection—SELEX (Systematic Evolution of Ligands by EXponential Enrichment). Shortly after this invention it was clear that these molecules can play a crucial role especially in medicine [122]. 

The term “aptamer” is a combination of two words the Latin *aptus*—meaning “*to fit*” and the Greek meros—meaning “*the part*”. Aptamers are single stranded DNA or RNA oligonucleotides composed of typically 80 to 100 bases [123,124]. The aptamers, in respect, of specificity to their target, are to a certain analogue the antibodies and can be considered as artificial/chemical antibodies. However, in the cells, short RNA sequences (tRNA) are used for delivery of the amino acids to the ribosome for the purpose of peptide synthesis. Thus, the naturally occurring aptamers represent an important step in molecular evolution. However, in contrast with antibodies aptamers, especially after their chemical modifications, are more stable. They can be heated up to 95 °C without loss of their structure after cooling and can be stored in dry conditions in the deep freezer (−18 °C) for up to one year. They are selected in vitro, without using experimental animals. Therefore, the SELEX can be applied even for selection aptamers against toxins, pesticides, herbicides, pathogenic bacteria, or viruses. In addition, aptamers are cheaper in comparison to antibodies and practically do not cause any immune response in the body. Once the aptamer sequence is developed, it can be reproduced with high accuracy by polymerase chain reaction method (PCR) or by standard nucleic acid synthesis assay. Table 1 compares properties of aptamers and antibodies. Thus, aptamers can serve as a potential replacement of antibodies in diagnostic methods, for example in Enzyme Linked ImmunoSorbent Assay (ELISA) and in the therapy. Aptamers are also more stable in the harsh tumor environment in comparison with antibodies. Due to their low molecular weight (~8–25 kDa), aptamers penetrate the tissue more readily and faster in comparison with antibodies [125]. Aptamers can be also regenerated after denaturation, returning to their original functional shape with the same specificity. One of the disadvantages of aptamers is their nuclease sensitivity, which may cause problems in therapeutic applications [126]. However, chemical modifications of aptamers can substantially improve their lifetime in the body fluids. Aptamer resistance to endonuclease cleavage in a bloodstream can be substantially improved by their conjugation with nanoparticles or nano/micromotors [127,128,129]. In general, one of the advantages of DNA/RNA aptamers is the possibility of their wide spectrum of modifications that improve their binding properties and specificity. The specificity of DNA and RNA aptamers is similar, however, after chemical modification, RNA aptamers are better suited for the transport of substances into the cells because they more easily cross the membrane. Nevertheless, they are less stable in comparison with DNA aptamers [130].

The number of articles that report on the application of aptamers in the various fields of biomedical research increase substantially. In particularly, substantial focus is on the development of aptamer-based biosensors for diagnostics as well as for the improvement of the efficiency of the therapy by the targeted transport of drugs using nanoparticles modified by aptamers. Special attention is on the application of aptamers in the diagnostics and therapy of cancer (see for example Giudice et al. [1] for recent review of application of aptamers in the therapy and diagnosis of leukemia). Aptamers can be chemically modified by fluorescent probes or redox markers, by thiol, amino groups or by biotin, which increase their stability and enable for their immobilization at surfaces or at nanoparticles. Fluorescent and redox probes allow for the application of various methods for the detection of aptamer-target interactions.

One of the key capabilities of aptamers is changing their 3D structure under certain physical and chemical conditions. This change is induced by the environment as well as by their targeting molecules. The specific molecule (ligand) induces formation or stabilization of the structure of the aptamer binding site. Typical 3D structures include the stem, loop, triplex/quadruplex, or hairpin [131]. There are several types of interactions between aptamer and ligand such as hydrogen bonds, hydrophobic/electrostatic interactions, Van der Waals interactions or aromatic stacking [132,133]. The dissociation constant is usually in the pico–nanomolar range [125]. The scheme of aptamer folding and interaction with target is presented on Figure 7.

Up to now, aptamers have been developed for various targets such as thrombin, HBV virus, *E. Coli*, and other bacteria, aflatoxin B1 or M1, PSA, CEA tumor markers, or cell membrane proteins such as protein tyrosine kinase 7 (PTK7) in the CCRF-CEM T-lymphoblast membrane and others [134,135,136,137,138,139,140,141]. The spectrum of aptamer targets is practically unlimited. In the case of cancer, they can recognize oncoproteins, cancer markers and metabolites associated with cancer processes [142].

In practice, great effort has been made in the selection of aptamers for cancer cells that target the oncomarkers in their membranes. A method capable of recognizing differences between healthy cells and the cells that are altered or damaged by the cancer is of crucial importance. Aptamers have the potential to identify these cells with minimal effect on the normal cells [143]. Aptamers are developed by a combinatory chemistry method SELEX, which was discovered in 1990 independently by three teams [144,145,146]. In this method, random sequences of DNA libraries are prepared by automated DNA synthesis. The size of a randomized region can vary from 30 to 60 nucleotides, flanked at both sides with a specific DNA sequence for PCR amplification. The theoretical diversity of individual oligonucleotides in these libraries is rather large. For example, in the case of oligonucleotides composed of 40 bases, it is 4^40^ = 1.2 × 10^24^. In practice, however, a considerably smaller library of approximately 10^13^ to 10^15^ molecules is used [147]. This library is incubated with the target molecules. During incubation, certain specific sequences of oligonucleotides bind to the target molecules. Subsequently, unbound sequences are removed, and the bound sequences are separated from the target molecules by special methods. The separated sequences are amplified and used for the next cycle of SELEX. Depending on the requirements, there are a necessary 10 to 20 SELEX cycles for selecting the aptamer with high specificity to the target. However, each additional cycle yields oligonucleotide sequences (aptamers) with higher affinity to target molecules than in the previous generation [128]. Currently, there are several modifications of SELEX that increase its efficiency, thereby reducing the number of cycles required to obtain sufficiently specific aptamers. These include, for example, negative SELEX, which eliminates the possibility of binding non-specific aptamers to the immobilization matrix, capillary electrophoretic (CE) SELEX, which separates specific sequences from non-specific differences in electrophoretic mobility, Cell-SELEX, which selects aptamers for cell membrane SELEX proteins and many other types of SELEX to reduce the number of cycles down to 1 to 2 [148].

### 4.1. Cell-SELEX

Cell-SELEX is important for the development of aptamers to tumor markers (Figure 8). This is a SELEX modification that develops aptamers specific for cancer cell membrane proteins in their natural environment. It is an alternative to protein-SELEX, which is traditionally used to select aptamers for proteins or other molecules. Cell-SELEX has the advantage that it needs no separation of the purified proteins and allows for the development of aptamers for proteins at the site of their natural occurrence [149,150].

Cell-SELEX starts similarly to the conventional SELEX by generating a library of DNA/RNA sequences. This is followed by the incubation of nucleic acids with target cells and removal of unbound sequences. Bound sequences are eluted from the cell membrane proteins by heating to 95 °C. An important step is the incubation of the obtained aptamers with negative cells, thereby removing sequences with lower specificity. The DNA/RNA sequences obtained are amplified by PCR. If more specific aptamers are needed, the Cell-SELEX process can be repeated several times. Each generation of aptamers has more optimal binding properties than the previous ones [151].

### 4.2. Aptamers for Cancer Markers

Detection of tumor markers is expensive and time consuming, with traditionally used clinical methods such as immunohistochemistry, which detect specific antigens by antibodies or flow cytometry, which is mainly used for immunophenotyping. The disadvantages of these and other diagnostic methods have initiated research focused on the development of new, more rapid, precise, and cheaper methods for oncomarkers detection at a time when their levels in body fluids are in relatively harmless concentrations.

In 2008, Shangguan et al. [141] published a work where they developed a new strategy for the detection of tumor markers using the Cell-SELEX method. By applying Cell SELEX to leukemia cells, they found an aptamer that bound to CCRF-CEM cells, so-called T-lymphoblasts of acute lymphoblastic leukemia (ALL). In this approach, from a large number (10^15^) of DNA sequences, a group was selected, named sgc8, which showed affinity for both lymphoblastic and myeloblastic leukemia cells. After 20 cycles of SELEX, they found a sequence with optimal binding properties to CCRF-CEM cells. Of the sgc8 family of aptamers, the sgc8c aptamer has been shown to have the highest affinity for these cells. The membrane protein responsible for binding to the aptamer was identified. It was a protein tyrosine kinase 7 (PTK7), which was confirmed by both, gel electrophoresis and flow cytometry. Aptamers for various cancer cells and molecules associated with cancer have been identified in a similar way. Some of them, for various types of cancers, are described in Table 2.

Several aptamers are currently in various stages of clinical trials for therapy of hematological [1] and pancreatic cancer [177] diseases. Pi et al. [178] also reported preparation of RNA/DNA hybrid nanoparticles that contained DNA aptamer specific to annexin A2 in ovarian cancer cells. The targeted delivery of doxorubicin (DOX) into the cancer cells was provided by CG rich sequences in the nanoparticles into which DOX intercalate. It has been shown that nanoparticles enhanced the therapeutic effect of DOX in ovarian cancer cells. 

### 4.3. Modification of Polymer Nanoparticles by Nucleic Acid Aptamers and Antibodies

Surface functionalization of polymer nanoparticles is a crucial step to ensure specific targeted delivery of therapeutics to the desired site of pathologies. It is possible to achieve surface functionalization of polymer nanoparticles by various strategies. Functionalization by antibodies can be achieved by adsorption, covalent binding or through adapter molecules. It is, however, important to provide optimal surface density and correct orientation of antibodies. Depending on the method of conjugation, the immobilization of antibodies can be random or oriented. Out of all the possible orientations, the, end-on orientation is the most common [179]. 

The simplest method of immobilization of antibodies at surfaces is physical non-covalent adsorption. This adsorption is stabilized by ionic bonds between oppositely charged surfaces of antibodies and nanoparticles. Physical adsorption includes weak interactions such as electrostatic interactions, hydrogen bonds, hydrophobic forces and van der Waals forces [179]. In the study of Choi et al. [180], three surface modification methods of polymer nanoparticles were reviewed: adsorption method, charged adsorption method, and bio-conjugation. The prepared docetaxel loaded poly(lactic-*co*-glycolic acid) (PLGA) nanoparticles were functionalized with antibody Herceptin^®^. Based on these results, the bioconjugation proved to be more efficient in terms of stability. The least stable solution proved to be the adsorption method [180]. In addition, the adsorption method unfortunately needs a high concentration of expensive antibodies [179]. Covalent methods require first the activation of the surface of the nanoparticles and based on the demand, chemical modification of the antibody can be necessary. Covalent methods involve carbodiimide chemistry, maleimide chemistry, and click chemistry. These methods have proven to have higher stability and efficient reproducibility. In the study of Xu et al. [181] a poly(ethylene glycol)-b-poly(ε-caprolactone) (PEG-PCL) docetaxel loaded nanoparticle complex was reported. This nanoparticle system was functionalized with programmed death ligand 1 antibodies (PD-L1) by carbodiimide chemistry and later the cytotoxic effects were studied on gastric cancer cell lines HGC27, MGC803 and MKN45. The results have shown the high efficiency of functionalized nanoparticles in comparison with non-functionalized against gastric cancer cells [181]. In Table 3, the selected examples of antibodies and their possible therapeutic use in nanocarrier-drug complexes are presented.

Currently the more often methods used for aptamer conjugation to the surface of polymer nanoparticles are: carbodiimide method, thiol-maleimide chemistry or avidin-biotin coupling. Carbodiimide method is frequently used in bioconjugation for drug delivery. For adsorption, 1-ethyl-3-(3-(dimethylaminopropyl)-carbodiimide (EDC) compound is often used. In this case, EDC acts as an activator of carboxylic residues to react with amino groups on ligands, producing covalent amide bond via carbodiimide coupling [195]. For the functionalization of PNPs, carboxylation of hydrophilic part is often needed, producing a bonding area for 5′-amino-aptamer [196].

Avidin-biotin coupling (Figure 9) can be obtained through Van der Waals forces and hydrogen bonds. Avidin-biotin bond is known to be the strongest, non-covalent bond with a dissociation constant of ≈10^−15^ M [195]. In the study of Ninomiya et al. [197] biotinylated aptamers were conjugated on avidin-treated liposomes. Thermosensitive doxorubicin-loaded poly(NIPMAM-*co*-NIPAM) nanoparticles were studied against breast cancer cell lines. The aptamer functionalized liposomes have proved to be efficient against chosen breast cancer cell lines.

Thiol-maleimide chemistry is commonly used in the functionalization of polymer nanoparticles with thiolated therapeutics or targeting ligands that bind to maleimide compounds incorporated in the nanoparticles [198]. For example, Alibolandi et al. [199] presented camptothecin-loaded pegylated PAMAM dendrimer, functionalized with AS1411 anti-nucleolin-aptamer by thiol-maleimide chemistry for targeting colorectal cancer cells.

The above discussed examples of preparation nanoparticles for targeted drug delivery were based on the modification of these nanostructures by aptamers or antibodies. However, also molecularly imprinted silicon nanoparticles were reported that can recognize the cancer markers at the surface of the cells. The scheme of preparation of molecularly imprinted silicon nanoparticles is presented on Figure 10 and the method of preparation is published in detail by Piletska et al. [200]. 

Briefly, 12 nm diameter silica nanoparticles were incubated in the phosphate buffer in the presence of the solid-phase. Phosphate ions were used as a catalyst in the ripening process for providing growth of larger particles. Material deposited in the vicinity of template molecules resulted in the formation of sol-gel molecular imprints after around 2 h. Selective washing and elution allows for isolation of the higher affinity nanoparticles. The authors demonstrated preparation of these nanoparticles against various targets such as melamine, vancomycin and trypsin and approved their high affinity using enzyme linked immunosorbent assay (ELISA) with sub nM detection sensitivity. 

### 4.4. Methods of Immobilization of Antibodies and Aptamers at the Surface of Nanomotors

Nano/micromotors offer novel potential platforms for sensing and as carriers for drug delivery. Surface modification of nano/micromotors with different components is very important for further applications. Therefore, functionalization chemistry is used to change the surface properties, such as charge, hydrophobicity, or hydrophilicity, which affect toxicity, biodistribution, and circulating time in biological systems [201]. PEG is commonly used to increase stability for in vivo application and to prevent degradation by the immune system [202]. The size, shape, charge, and surface functionalization have a considerable impact on the cellular uptake [203]. Aptamers and antibodies can serve as recognition elements in biosensing or in targeted delivery. Both probes can be immobilized at the surface of nano/micromotors though covalent and noncovalent binding such as EDC-NHS crosslinking, thiol-metal interaction, avidin-biotin interaction, or π-interactions [204], concerning the type of materials, purpose, and efficiency as for antibodies [205], as well as for aptamers [206] (Figure 11). In comparison with nanoparticles, the influence of the fuel or an external source for movement must be also considered due to the possible effect on the surface properties.

Antibodies and aptamers were immobilized on nano/micromotors depending on the sensing strategy or targeted delivery design. For example, for graphene-based nanomotors, the aptamers were immobilized on the surface through non-covalent π-π stacking between DNA and graphene oxide (GO) [102,207]. Thiol-metal interaction was used for immobilization of thiol modified aptamers on AuNPs in nanocomposite nanomotors [208]. Carbodiimide chemistry was applied for antibody functionalization. An antibody for *S. Aureus* loading was immobilized on Au-Ni-Au nanowires. The nanowires were first coated with thiol self-assembly monolayer by overnight immersion in a mixture of mercaptoundecanoic acid (MUA) and mercaptohexanol (MCH). Then, carboxylic groups were activated through EDC-NHS chemistry and conjugated with an antibody. To reduce non-specific binding, the remaining amine reactive-esters of the activated thiol monolayer were blocked with ethanolamine [101] The EDC-NHS coupling was used for the immobilization of VCAM-1 polyclonal antibody on platelet-derived nanomotors [209]. Another approach used modification through thiol terminated PEG linker that allowed for coupling of the targeting moiety antibody [210].

## 5. Polymer Nanoparticles Modified by Aptamers and Antibodies in Targeted Therapy of Cancer

For the last two decades, researchers have been working on the development of the nanocarrier systems for targeted drug delivery. Until now, there have been presented plenty of designs with various types of nanoparticles, including polymer nanoparticles. Polymer nanoparticles represent an attractive tool for the diagnosis and treatment of a wide range of diseases. Advances in controlled polymerization have enabled the development of multifunctional polymeric nanoparticles with precise control of architecture, shape, size, surface charge, and functionalization [211]. Macromolecular chemistry allows for the modification of nanoparticles without altering their physical, chemical, and biological properties [212].

Nowadays, the effort is focused on the development of more accurate targeting methods for the treatment of various diseases, including cancer. The targeting can be achieved by aptamers and antibodies, which specifically bind to surface antigens or proteins presented on the pathological cells or tissues [213].

### 5.1. Polymer Nanoparticles Modified by Antibodies

In this part, we present several examples of recent works on the development of polymer nanoparticles (PNPs) modified by antibodies for targeted drug delivery. In the study of Patel et al. [214], it was proposed a design of polymer nanoparticle-antibody complex consisting of poly(lactide-*co*-glycolide) (PLGA) nanoparticles conjugated with anti-EGFR (epidermal growth factor receptor) antibodies that target the lung cancer cells. As an active drug agent, cetuximab was immobilized to docetaxel loaded PLGA PNPs. The size distribution of these PNPs was 128.4 ± 3.6 nm with conjugation efficiency of almost 40%. In vitro studies demonstrated high anti-proliferative activity of these PNPs with sustained drug release of 25% after 48 h at pH 5.5. In vivo studies revealed immersive reduction in the growth of the tumor [214].

Canakci et al. [215] reported a polymer nanogel with immobilized antibodies anti-CD4, specific for CD4^+^ T lymphocyte cells and CD4^high^ T cell lymphoma as well. The nanogel formation presented in this paper is defined as chemically cross-linked, water soluble polymeric nanoparticle. The nanogel was loaded with cytotoxic drug mertansine (DM1), which effects the inhibition of microtubules. The drug loaded polymer nanogel-antibody complex has proven to be an effective targeting and drug delivery agent.

In the study of Kuo and Tsai [216] an interesting complex consisting of rosmarinic acid (RA) and curcumin (CUR) loaded polyacrylamide-cardiolipin-poly(lactide-*co*-glycolide) nanoparticles (PAAM-CL-PLGA) with conjugated 83-14 monoclonal antibody (83-14 MAb) has been reported. The PNPs successfully modulated the endocytosis pathway trough insulin receptors and activated receptor mediated transcytosis across the blood brain barrier, which made it possible to deliver the loaded drugs. It has been shown that developed PNPs revealed neuroprotective abilities due to antioxidative properties of the loaded drugs RA and CUR. This can prevent neurodegeneration during the treatment of Alzheimer’s disease. The PNPs complex can be useful in other applications for brain related neurodegenerative diseases as well.

### 5.2. Polymer Nanoparticles Modified by Aptamers

As we have shown in part 4, DNA/RNA aptamers are relatively new receptors that can improve the targeted delivery of chemotherapeutic drugs into the cancer cells. As it is followed from Table 2, a large number of DNA or RNA aptamers were developed for the recognition of various cancer markers at the surface or inside the cancer cells. The number of publications focused on the application of aptamers in targeted drug delivery substantially increases. This is in particularly due to the high flexibility of aptamers and possibility of directed modification at the surface of polymer PNPs that provide sufficient flexibility of aptamer binding site for onkomarker recognition. In this part, we will show several examples of the effectivity of aptamer-polymer PNPs conjugates for targeted drug delivery. 

In the paper by Kong et al. [217], it has been proposed PNPs consisting of star-shaped co-polymer cholic acid and poly(ε-caprolactone-ran-lactide) (CA-(PCL-ran-PLA)) modified by aptamer (AS1411) and by polydopamine (pD). AS1411 aptamer contains G-rich sequences that targets nucleolin at the surface of the cells, which is overexpressed in various cancers, e.g., breast, prostate and lung cancer, melanoma, glioma, etc. The PNPs have been loaded by docetaxel for chemo-photothermal therapy of breast cancer. In vivo and in vitro cytotoxicity and antitumor assays showed improved survival times as well as significantly reduced tumor cell proliferation. The effective target delivery and an exceptional therapeutic efficacy were due to synergistic chemo-photothermal effect.

Yang et al. [218] prepared a doxorubicin (DOX) loaded DNA aptamer functionalized photoresponsive hyperbranched polymer that can self-assemble into nanoparticles (HDNPs). In vitro studies proved specific binding, internalization of the delivery system and cytotoxicity against cancer cells. The PNPs revealed unique characteristics, such as rapid disassembly upon UV radiation, which released the internal content, as well as high stability and drug loading (32.35 ± 1.17 μg/mg). HDNPs and photoresponsive drug release of DOX caused significant inhibition of the cancer cell proliferation. In a recent study of Han et al. [219] the PNPs coated with exosomal membrane (EM-coated PLGA) functionalized with DNA aptamers AS1411 were reported (Figure 12). The EM coating was not affected by the aptamer modification and the proposed PNPs indicated good biocompatibility and efficient tumor targeting.

Gui et al. [220] developed polymer-lipid nanoparticles (PLNP) modified with CD133 aptamers for targeted delivery of all-trans retinoic acid (ATRA) (ATRA-PLNP-CD133) to osteosarcoma initiating cells (Figure 13). The ATRA-PLNP-CD133 PNPs were of 125 nm average size and revealed good encapsulation efficiency of 86.4%. The toxicity studies showed high antitumor efficacy of PNPs. Comparison of tumor volume inhibitory rates between pure ATRA, ATRA-PLNP, and ATRA-PLNP-CD133 resulted in 44.3%, 53.7%, and 81.1%, respectively. The experiments proved superior therapeutic efficacy of ATRA-PLNP-CD133, which includes 144 h long sustained release of retinoic acid.

In addition to lipid coated nanoparticles, those based on the coating of natural membranes are also explored. These biomimetic nanoparticles are composed of the core, which can be polymer, metal, or carbon-based material surrounded by natural membrane of red blood cells, cancer cells, neutrophil, platelet, macrophage, or stem cells. The coating is provided by various mechanisms, such as ultrasonication or extrusion as well as combination of extrusion with ultrasonication. The extrusion method, consisting in the preparation of the mixture of core nanoparticle and corresponding cell membrane with subsequent extrusion through the pore containing membranes of the diameter of 100 to 200 nm [221]. The biomimetic nanoparticles are of low cytotoxicity on normal cells. For example, biomimetic gold nanoparticles prepared from seaweed *Gelidium pusillum* were prepared and studied in respect of their cytotoxicity. It has been shown biocompatibility and neglected cytotoxicity of these biomimetic NPs [222]. 

Polymer nanoparticles and especially those based on biomimetic structures such as lipid membranes or as having the shell based on natural cell membranes are highly biocompatible and biodegradable. No toxicity was reported for biocompatible polymer-based nanoparticles [221]. 

## 6. Nanomotors Modified by Aptamers and Antibodies in Targeted Therapy of Cancer

The functionalization of nano/micromotors is crucial for biomedical applications. In contrast with PNPs, the advantage of nano/micromotors is in the possibility of enhancements of their movement to the target due to several forces that were discussed in part 3. Modification of nano/micromotors by active biological components or drugs allowing their usage for biosensing, diagnostics, therapy, or targeted drug delivery. Aptamers and antibodies can interact with their target molecules with high affinity and can be used as receptors in diagnostic and therapeutic tools [223,224]. The application of nanomotors in the early diagnosis of cancer is essential for improvement of patient survival. At the same time, the targeted drug delivery using nanomotors can improve the effectiveness of the therapy. The antibodies or aptamers at the surface of nanomotors recognize the cancer markers at the surface of the cancer cells and thus can interact preferably with affected tissues. This is crucial for avoiding the undesirable delivery of chemotherapeutics to the normal cells.

### 6.1. Nanomotors Modified by Antibodies

Receptor-functionalized nanomotors can also provide direct isolation of target molecules from biological samples, e.g., circulating tumor cells (CTC), which indicate various cancer diseases [225]. For example, human pancreatic cancer cell with overexpressed anti carcinoembryonic antigen (anti CEA) were isolated using Ti/Fe/Au/Pt microrockets functionalized with CEA monoclonal antibodies [226]. The functionalized microrockets powered by H_2_O_2_ successfully bound and isolated CEA positive cancer cells, and even dead CEA positive cell or their cellular fragments. 

The antibody modified catalytic nanomotors has shown success in targeted cancer treatment. In a recent study by Hortelao et al. [210], the antibody functionalized urease-powered nanomotors were demonstrated for targeting bladder cancer spheroid (Figure 14). The antibody binds to the fibroblast growth factor receptor 3 (FGF3) overexpressed in bladder cancer enabling targeting as well as therapeutic effect. Nanomotors were based on mesoporous silica nanoparticles conjugated with PEG and anti-FGF3 antibody (MSNP-Ur/PEG-Ab). Functionalized nanomotors have enhanced internalization by 14-fold compared with bare passive nanomotors. The therapeutic effect of functionalized nanomotors was confirmed by decreasing the viability of tumor spheroids. 

Biodegradable 3D printed microswimmers, which are hydrogel-based and magnetically powered, were used for the accomplishment for cargo delivery and release based on enzymatic sensing of matrix metalloproteinase 2. In addition, antibody anti-ErbB 2 conjugated magnetic nanoparticles were released for targeting SKBR3 breast cancer cells [227].

Tumor cells produce H_2_O_2_, which allows for the fueling of self-propelled nanomotors. Multicomponent self-propelled nanorobots with magnetic guidance were demonstrated for intracellular delivery of doxorubicin (DOX). The nanorobots based on multiwalled carbon nanotubes (MWCNTs) functionalized with anti-epithelial cell adhesion molecule antibody (anti-EpCAM mAb) and enriched by Fe_3_O_4_ particles have shown significant cytotoxicity effect against human colorectal carcinoma (HCT116) indicating efficiency of delivery compared to free DOX. Furthermore, deep penetration of nanorobots for in vivo application was verified using 3D HCT116 spheroids [228].

### 6.2. Nanomotors Modified by Aptamers

Using of aptamers due to their flexibility allowing application of various recognition strategies. Among them is the fluorescence switching turn OFF-ON strategy that is frequently used in biosensing. This approach is based on Förster (or fluorescence) resonance energy transfer (FRET) between fluorophore and quencher. Graphene oxide (GO) is commonly used as an efficient quencher for many fluorescent dyes e.g., fluorescein. GO can easily physically adsorb DNA or RNA through π-π stacking and hydrogen bond interactions [229]. Thus, fluorescently modified DNA or RNA can be adsorbed on the GO surface, resulting in quenched fluorescent signal. The presence of target leads to fluorescence recovery upon interaction of the probe and target. In particular, Esteban et al. [102] demonstrated intracellular detection of target miRNA21 associated with many diseases, e.g., cancer, using ultrasound-propelled gold nanowires (AuNWs) functionalized with fluorescently labeled ssDNA and the OFF-ON sensing strategy. AuNWs were modified with GO. The fluorescent signal increased in the presence of target molecule, due to releasing fluorescently labeled ssDNA from nanomotors and fluorescence recovery. The fluorescent signal was proportional to the ultrasound exposure time and the applied voltage. A similar approach of intracellular sensing was reported for the detection of E6 miRNA in the human papilloma virus associated with oropharyngeal cancer [230]. Subsequently, Beltrán-Gastélum et al. [17] also reported a similar sensing strategy for qualitative detection of cancer biomarker, known as amplified in breast cancer (AIB1), and frequently overexpressed in breast cancer (Figure 15). GO modified AuNWs were functionalized with fluorescein labeled aptamer specific to AIB1 (FAM-AIB1-apt) and propelled towards MCF7 cells. The higher fluorescence signal was observed in MCF7 cells compared to control normal HFF-1. The propulsion of nanomotors under the ultrasound field facilitates accelerated intracellular uptake and probe binding with the target molecule. Thus, the fluorescence intensity is greatly enhanced compared to static conditions (without ultrasound). Easier penetration through cell membranes allows the payload to be transferred faster.

The usage of aptamers for the isolation of various compounds including cells offers a cheaper alternative to antibodies. The in vitro isolation of human promyelocytic leukemia cells (HL-60) from a human serum was performed using aptamer modified self-propelled nanomotors [208]. Nanomotors were fabricated as nanocomposites of manganese oxide, polyethyleneimine, nickel, and gold particles. KH1C1 aptamer was attached through thiol group on gold part of nanomotor. The movement of nanomotors was based on catalytic decomposition of H_2_O_2_ on manganese dioxide via oxygen bubbles. The determination of HL-60 cells concentration was performed using electrochemical method. The performed studies did not reveal significant cytotoxicity of nano/micromotors on healthy cells. This includes fuel driven nanomotors using natural fuels, such as H_2_O_2_ or glucose as well [82]. In addition, the effect of ultrasound on the viability of the cells was not significant [78].

### 6.3. Application of Polymer Nanoprticles and Nano/micromotors in Controlled Drug Release

The development of an intelligent drug-delivery system with controlled and stimuli-responsive release has received tremendous attention. The effort is focused on the elimination of drug biotoxicity and the side effects to healthy cells and tissues. Drug release can be performed by several chemical and biological mechanisms such as dissolution, diffusion, osmosis, partitioning, swelling, erosion, and targeting. Stimuli-response drug release system can be based on different triggers such as microenvironments of cells or external sources (light, ultrasound, magnetic, heat) [231,232,233]. Cancer cells differ from normal cells, therefore these distinct features e.g., metabolic activity, growing paths, division, pH environment, can provide unique tools for a controlled drug release. Biodegradable polymeric nanoparticles with a stimuli response mechanism such as potential drug delivery platform have a big impact on the therapeutic efficacy of in vivo application [234]. Metal organic frameworks can also be used for this purpose. The synthesis of doxorubicin-loaded metal–organic framework nanoparticles (NMOFs) coated with a stimuli-responsive nucleic acid-based polyacrylamide hydrogel has been recently developed in Willner’s laboratory [23]. The basic principle of preparation and functioning of these nanoparticles is presented in Figure 16. The formation of the hydrogel is stimulated by the crosslinking of two polyacrylamide chains, P_A_ and P_B_, which are functionalized with two nucleic acid hairpins (4) and (5), using the strand-induced hybridization chain reaction. The resulting duplex-bridged polyacrylamide hydrogel includes the anti-ATP (adenosine triphosphate) aptamer sequence in a caged configuration. The drug encapsulated in the NMOFs is locked by the hydrogel coating. In the presence of ATP, which is overexpressed in cancer cells, the hydrogel coating is degraded via the formation of the ATP–aptamer complex, resulting in the release of doxorubicin. In addition to the introduction of a general means to synthesize drug-loaded stimuli-responsive nucleic acid based on polyacrylamide hydrogel-coated NMOFs hybrids, the functionalized NMOFs resolve significant limitations associated with the nucleic acid-gated drug-loaded NMOFs. The study reveals substantially higher loading of the drug in the hydrogel-coated NMOFs as compared to the nucleic acid-gated NMOFs and overcomes the nonspecific leakage of the drug observed with the nucleic-acid-protected NMOFs. The doxorubicin loaded, ATP-responsive, hydrogel-coated NMOFs reveal selective and effective cytotoxicity toward MDA-MB-231 breast cancer cells, as compared to normal MCF-10A epithelial breast cells. 

Moreover, nano/micromotors as a new generation of drug carriers offer many advantages, such as rapid transport, high tissue penetration, and controlled guidance. [235]. For example, self-propelled bowl-shaped stomacyte nanomotors based on biodegradable polymers PEG-b-PCL and PEG-b-PS and encapsulated PtNPs and propulsion by H_2_O_2_ produced in tumor cells were demonstrated as delivery system [236]. Degradation of nanomotors enabled a controlled drug release due to pH change. The same group presented redox-responsive nanomotor with glutathione as a trigger for drug release [237]. Glutathione has been found in the intracellular matrix in higher concentration compared with extracellular. Stomatocyte bowl-shaped nanomotors were based on block copolymer PEG-SS-PS and the polymer PEG-b-PS with incorporated disulfide bonds, and encapsulated PtNPs.

Despite the progress in the development of smart delivery system, there are many challenges for controlled and efficient drug release in human body. 

## 7. Comparison of Efficiency of Aptamer and Antibody-Based Targeted Therapy: Future Perspectives

The ultimate goal of pharmacology since 1900, when Paul Ehrlich stated the idea of the “magic bullet”, is to effectively target pathogens or unhealthy cells in the body without affecting healthy tissues. Until now, there are two known strategies for targeting pathologies [214]. The first major breakthrough was attributed to antibodies in 1975, thanks to Georges Kohler and Cesar Milstein. Only a year later, Ron Levy discovered monoclonal antibodies specific for cancer cells [238]. Years of research have encountered various limits including conjugation chemistry, manufacturing issues, product heterogeneity, cancer cell specificity and tumor penetration. [239]. For these reasons, new targeting elements—the nucleic acid aptamers—have been discovered in the 1990s, independently by three groups of researchers as it has been explained in part 4. In comparison to commonly used antibodies, aptamers can offer higher affinity and specificity to target cells as well as various advantages over antibodies, such as rapid production, low-cost, absence of immunogenicity, and thermostability. It is possible to produce them for a wide range of targets, such as small molecules, cells, bacteria, proteins, viruses, or tissues [240]. 

In the past, antibody therapeutics faced various challenges in clinical use. Some patients exhibited a strong immune response to the antibodies; often, occurrence was rapid clearance from the body by immune system, which required repeated administration. Low recognition of cell surface receptors in humans occurred as well [241]. In recent years, visible advances were made in the antibody design, although the high cost of development and high antibody-based drug prices to consumers remain a challenge. 

Antibody drugs have made a large impact in cancer therapies, however, passing the blood-brain barrier and clearance from the brain has hindered their use for the treatment of neurological diseases [242]. Aptamers are smaller than antibodies, approximately by one tenth; they are non-immunogenic and can across through tissue and cell membranes easier [125,147]. There are several animal studies where aptamer-drug complexes were able to successfully cross the blood-brain barrier and delivered the effective treatment. 

Monaco et al. [243] presented an aptamer-drug polymer nanoparticle complex able to penetrate the blood-brain barrier for glioblastoma treatment. The nanoparticle consisted of a copolymer poly(lactic-*co*-glycolic)-block-polyethylene glycol (PLGA-b-PEG). The platelet-derived growth factor receptors both α and β (PDGFRα/β) were recognized by DNA aptamers on resistant glioblastomas and tumorigenic glioblastoma stem cells. The drug was incorporated in the hydrophobic core of biodegradable PNPs and aptamers were conjugated on the surface of PNPs. The results showed higher entrapment efficiency of low solubility drug in the PNPs, increasing its bioavailability, which resulted in 1000-fold increase in cytotoxicity in comparison with the free drug. In conclusion, aptamer-PNPs have evidently crossed the blood-brain barrier and were able to accumulate at the tumor site. In vivo studies proved the specificity of selected aptamers and specific glioblastoma tumor uptake was observed. Following five days of the aptamer-PNPs intravenous administration, reduction in the expression of tumor markers was observed in tumor-bearing mice. 

The safety of application of nanostructures in targeted drug delivery and diagnostics is of crucial important for personalized medicine. In this case, the biomimetic nanoparticles are of substantial interest. They can be prepared by coating polymer nanoparticles or nanomotors by lipid membrane or even by membrane from the cells isolated from the blood of the patient. For recent achievements, please see the review by Jha et al. [221]. 

The important issue is also connecting with proper model for testing the nanostructures in targeted drug delivery. Despite over 30 years of investigations in this direction, the application of nanoparticles in clinical practice is still in a premature stage. The mice model for in vivo laboratory tests of nanostructures only partially fulfill the required conditions. Certainly, the host growth, metabolic rate, and host life of tumor in mice is different from those in humans. Therefore, the conclusion from the testing of nanomaterials in the chemotherapy of tumors in mice should not be automatically applicable on humans. Currently, there are very limited clinical trials of nanoparticles for the treatment of tumors. Mostly silica NPs, liposomes, lipid-based NPs, carbon NPs, iron oxide NPs and recombinant protein NPs have been included in clinical trials (see recent review by Tan et al. [21]). The future progress in this direction requires close collaboration of the researchers and clinicians. More focus should also be on the application of the nucleic acid aptamers in targeted drug delivery including stimuli responsive drug release. The clinical trials of this therapeutic method were not started yet. This holds true as well for nano/micromotors, for which the trials only in laboratory conditions were performed so far. 

## 8. Conclusions

The overview of recent works on polymer nanoparticles as well as nano/micromotors modified by antibodies or nucleic acid aptamers revealed a high advantage of these systems for targeted drug delivery in comparison with traditional chemo- or immunotherapy. The modification of nanocarriers by antibodies or nucleic acid aptamers can recognize the cancer markers at the surface or inside the cancer cells and can transport the chemotherapeutic drugs or antisense RNA inside the cells for blocking their proliferation. This approach minimizes the undesirable toxicity of drugs on normal cells. From the comparison of efficiency of antibodies and nucleic acid aptamers, it follows that DNA/RNA aptamers can be considered as rather perspective receptors in carrier systems due to their small size, non-immunogenicity, and stability. The recent research confirmed high advantage of nucleic acid aptamers in comparison with antibodies also in respect of their possibility to cross the blood-brain barrier. In addition, immobilization of aptamers at the surface of nanoparticles and nano/micromotors is easier and offers more possibilities in comparison with antibodies. Another advantage is the higher stability of aptamers-nanostructure conjugates, which is important for clinical applications. Among polymer nanoparticles, those based on biomimetic structures such as lipid membranes or even cell membranes are preferable in respect of biocompatibility and non-toxicity on normal, healthy cells.

Nanoparticles and nano/micromotors can both be effectively used for targeted delivery of chemotherapeutics. The advantage of nano/micromotors is in the controlled direction of diffusion process as well as acceleration of the movement. The various mechanisms of driving forces of nanomotors evidence that especially those based on ultrasound, magnetic field, or hybrid ultrasound-magnetic field mechanisms are preferable.

A revolutionary new mechanism of targeted drug delivery based on stimuli responsive drug release in the tumor cells is rather promising. However, the research in this direction is still in the beginning and in a level of laboratory testing. At the same time, only limited clinical trials are currently available for nanoparticle cargos. The problem consists in the proper selection of animal models as well. As it has been mentioned above, the murine model of cancer does not exactly correspond to those of humans. So far, there is no clear understanding of the molecular mechanisms of transformation of the normal cells into the tumor cells as well as on the mechanism of tumor progression. It is also not clear how nanomaterials interact with tumor cells, and how they circulate and penetrate cancer cell membranes. Therefore, progress in the application of nanostructures in theranostics will depend on close collaboration of researchers with clinicians.

Further effort is also required for the development of optimal chemical modification of aptamers that improves their stability in complex biological fluids and for maintaining their binding site for efficient recognition of target molecules. 

Comparative analysis of the effectivity of polymer nanoparticles and nanomotors as targeted delivery systems has not yet been performed on identical cell targets. However, it is likely that nano/micromotors represent a perspective delivery system thanks to the additional enhancement of drug transport driven by various physical or chemical forces.

## Figures and Tables

**Figure 1 polymers-13-00341-f001:**
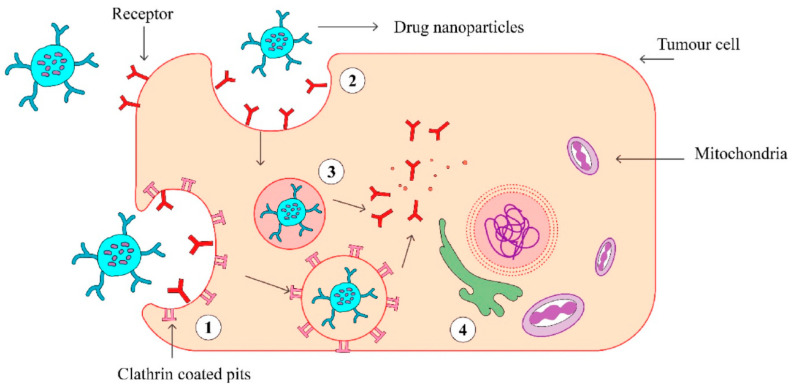
The scheme of endocytosis of the nanoparticle modified by antibodies that recognize cancer markers at the cell membrane. (**1**) Receptor (aptamer or antibody) on the drug-loaded nanoparticles recognize the clathrin-coated pits in the tumor cell and bind to it. (**2**) Phagocytosis of drug-loaded nanoparticles facilitates the transport of the carrier into the tumor cells. (**3**) An endocytic vesicle is formed. (**4**) Endosome induces the release of drug from the nanoparticles and penetrates into the nucleus. Reproduced from [21] with the permission of Elsevier.

**Figure 2 polymers-13-00341-f002:**
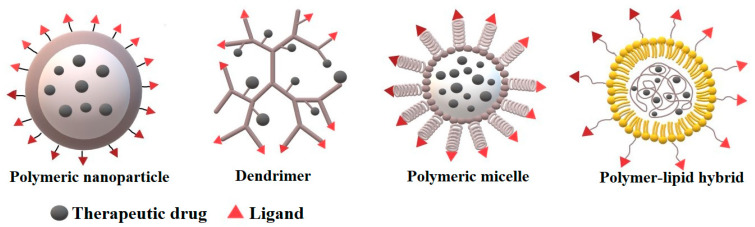
The different types of polymer nanocarriers for drug delivery.

**Figure 3 polymers-13-00341-f003:**
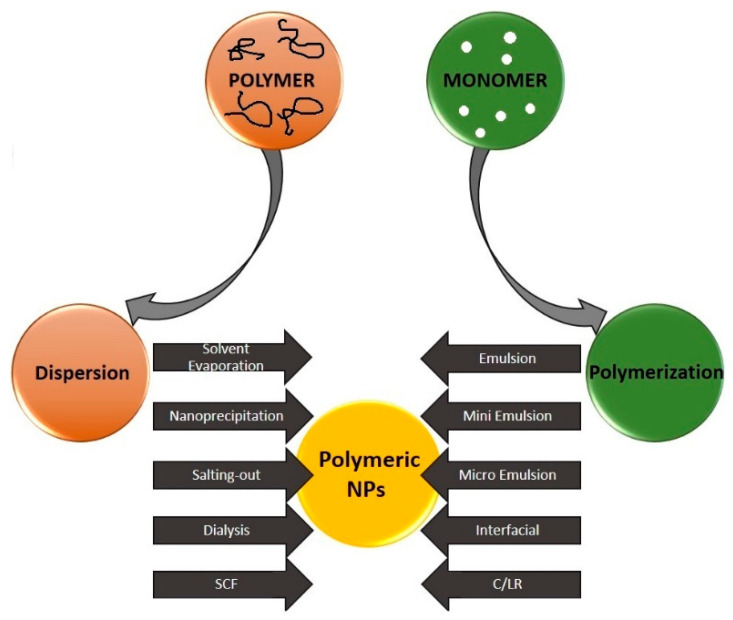
Schematic representation of various techniques for the preparation of polymer nanoparticles. SCF: supercritical fluid technology, C/LR: con-trolled/living radical. Reproduced from Rao and Geckeler [32] with permission of Elsevier.

**Figure 4 polymers-13-00341-f004:**
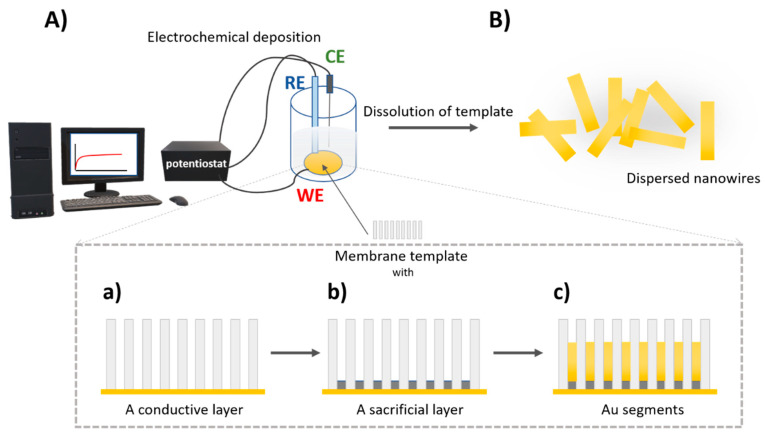
The scheme of preparation of nanowires using a membrane template-assisted electrodeposition (**A**) the three-electrode electrochemical cell using a membrane with a conductive layer as a working electrode (WE), RE—reference electrode, CE—counter electrode. Detailed view of electrodeposition process in a membrane (**a**) a sputtered conductive layer e.g., Au, (**b**) electrodeposition of a sacrificial layer, (**c**) electrodeposition of nanowire Au segment; (**B**) removal of a sacrificial layer and membrane resulting in free nanowires in water.

**Figure 5 polymers-13-00341-f005:**
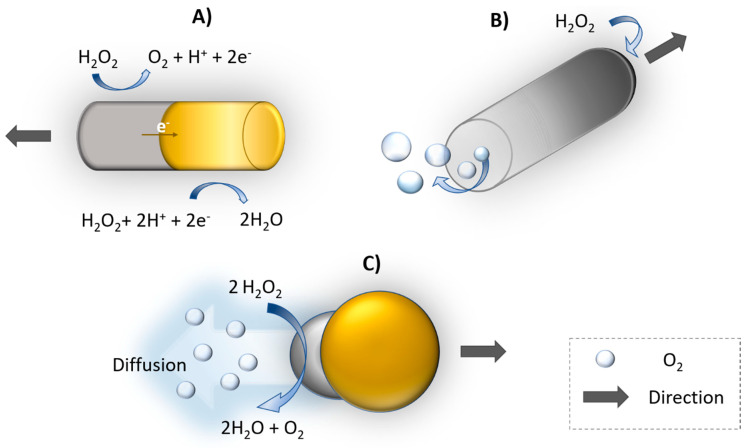
Schematic illustration of propulsion mechanisms of catalytically propelled nanomotors. (**A**) Self-electrophoresis, (**B**) bubble propulsion, (**C**) diffusiophoresis (example of composition Au-yellow, Pt-gray).

**Figure 7 polymers-13-00341-f007:**
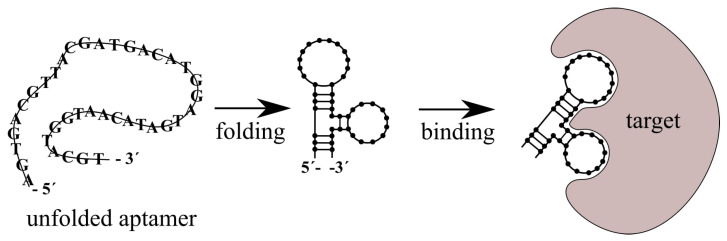
Scheme of transformation of aptamer in contact with target molecule. Aptamer forms 3D conformation (folding) and bind to the target.

**Figure 8 polymers-13-00341-f008:**
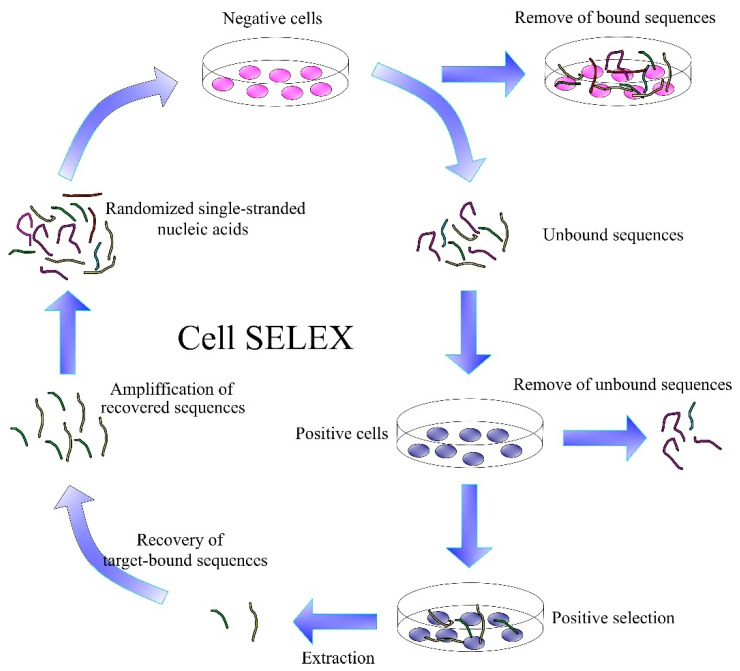
Scheme of the Cell SELEX.

**Figure 9 polymers-13-00341-f009:**
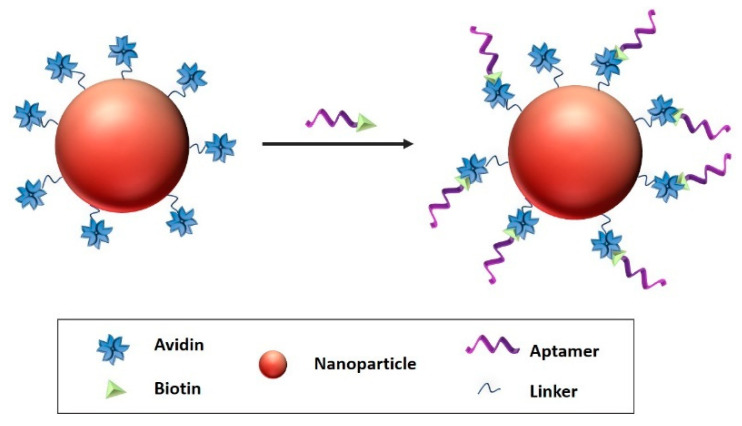
Avidin–biotin coupling, biotin attached the aptamer with avidin linked to the surface of the nanocarrier Reproduced from Odeh, et al. [198].

**Figure 10 polymers-13-00341-f010:**
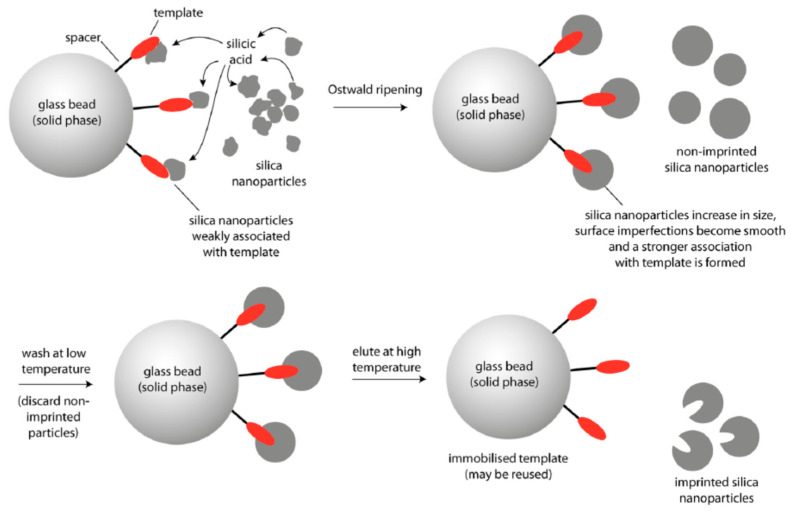
Schematic representation of the preparation of surface molecularly imprinted silica nanoparticles by the process of Ostwald ripening in the presence of an immobilized template [200].

**Figure 11 polymers-13-00341-f011:**
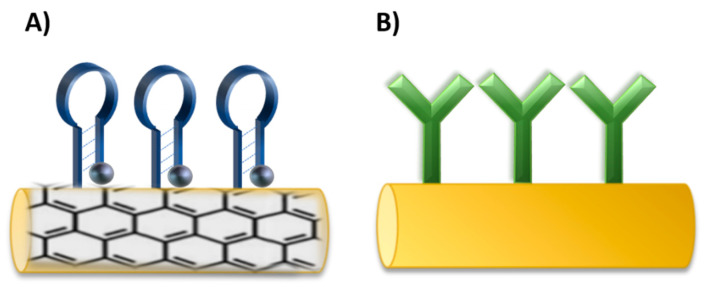
Schematic illustration of immobilization of (**A**) aptamers and (**B**) antibodies on the surface of nano/micromotors.

**Figure 12 polymers-13-00341-f012:**
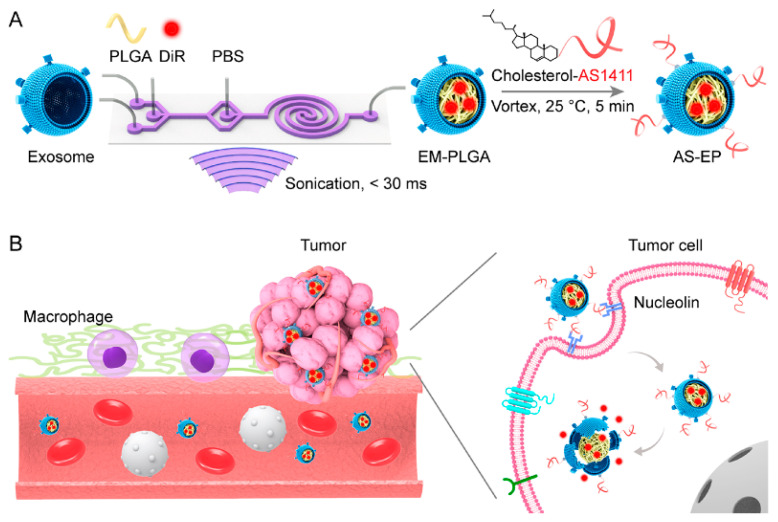
The scheme of PNPs composed of exosome shell (EM) and PLGA core and modified by AS1411 aptamers (AS-EP) for tumor targeting: (**A**) Scheme of preparation AS-EP; (**B**) scheme of AS-EP extended circulation in blood. The PNPs were characterized by reduced uptake by macrophages, and by enhanced targeting to tumor cells with overexpressed nucleolin. Reproduced from Han et al. [219] with permission of American Chemical Society.

**Figure 13 polymers-13-00341-f013:**
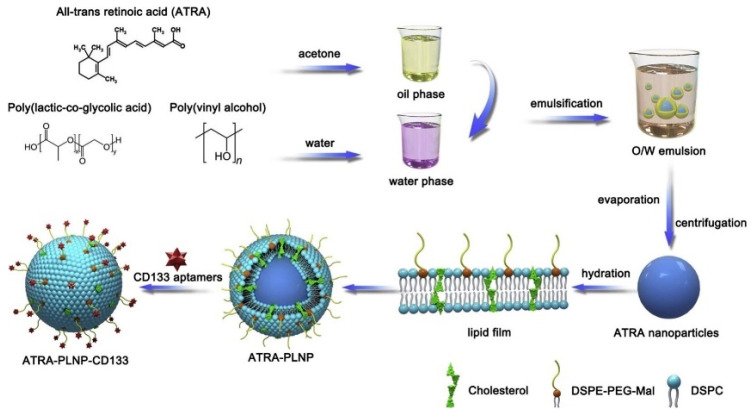
The scheme of preparation of ATRA-loaded lipid-polymer nanoparticles. The oil phase was formed by dissolving ATRA and PLGA in acetone. Then, the oil phase was slowly injected into poly(vinyl alcohol) (PVA) solution by homogenization. After the recovery by evaporation and centrifugation, the PNPs were hydrated with the lipid film. The CD133 aptamers were conjugated to the nanoparticles by the interaction of the maleimide on nanoparticles and the sulfhydryl groups of CD133 aptamers. Adopted from Gui et al. [220].

**Figure 14 polymers-13-00341-f014:**
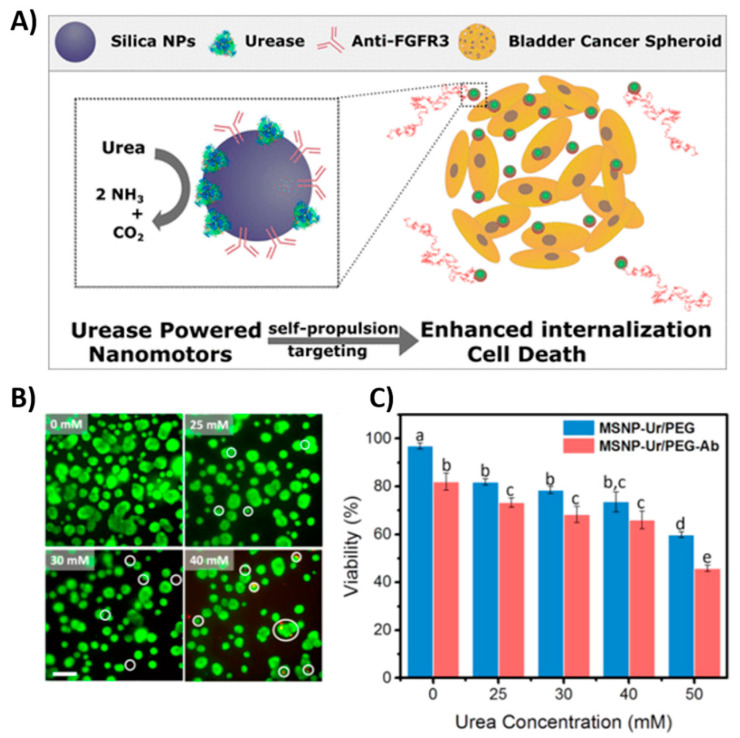
Targeting 3D bladder cancer spheroids with urease-powered nanomotors based on silica NPs. (**A**) Schematic illustration of targeting bladder cancer using antibody (Anti-FGR3) functionalized urease-powered silica NPs. (**B**) Live/dead assay of spheroids with nanomotors at 0-, 25-, 30-, and 40-mM urea after 4 h of incubation (scale bar 200 μm). (**C**) Quantification of spheroids’ viability after 4 h of incubation with urease/polyethylene glycol nanomotors (MSNP-Ur/PEG) (blue) and antibody-modified urease nanomotors (MSNP-Ur/PEG-Ab) (red) at different urea concentrations; different superscripts denote significant differences among groups with *p* < 0.05, *n* = 3, and results shown as mean ± SE Adapted from [210] with permission of the American Chemical Society.

**Figure 15 polymers-13-00341-f015:**
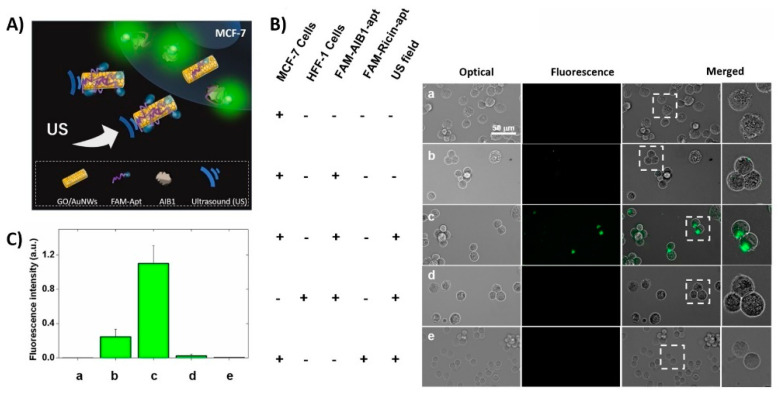
Intracellular fluorescent OFF-ON detection of AIB1 protein in MCF-7 breast cancer cells using US-propelled aptamer (FAM-AIB1-apt) functionalized nanomotors. (**A**) Schematic illustration of OFF-ON strategy based on fluorescence switching. Fluorescence of FAM-AIB1-apt is quenched due to adsorption at GO surface. Subsequently, the fluorescence signal is recovered in presence of specific target AIB 1. (**B**) Representative microscopic optical, fluorescence and merged images corresponding to cells treated under different conditions (**a**–**e** rows). (**C**) Bar plot displaying the fluorescence intensity obtained under different treatments (**a**–**e**) describe in (**B**). Ultrasound field parameters: voltage amplitude 2 V, resonant frequency 2.66 MHz, and 15 min. Scale bar 50 μm. Reproduced from [17] with permission of Wiley-VCH.

**Figure 16 polymers-13-00341-f016:**
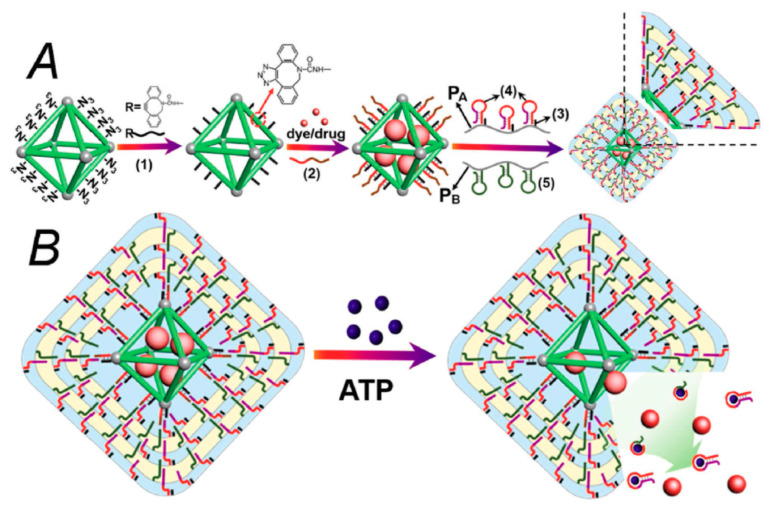
(**A**) Synthesis of ATP-responsive DNA/polyacrylamide-hydrogel-coated NMOFs loaded with a fluorescence dye or doxorubicin. (**B**) Schematic mechanism to unlock the hydrogel-coated NMOFs and to release the load via the formation of ATP–aptamer complexes. Adopted from Chen et al. [23] with the permission of Wiley-VCH.

**Table 1 polymers-13-00341-t001:** Comparison of the properties of antibodies and aptamers [128,129].

Features	Antibody	Aptamer
Specificity	High	High
Size	Relatively high	Small
Stability	Unstable	Stable
Affinity	High	High
Immunogenicity	High	No humoral response
Potential target	Immunogenic molecules	Any target
Production	In vivo	In vitro
Cost	Expensive	Relatively cheap
Modification	Limited	Almost unlimited
Time to generate	~6 months	~3–7 weeks
Renal separation	Slow	Fast

**Table 2 polymers-13-00341-t002:** DNA/RNA aptamers to various types of cancer.

Target	Aptamer	Aptamer Sequence 5′ → 3′	K_D_, nM	Ref.
		**Leukemia**		
PTK7 (ALL)	Sgc8c	ATC TAA CTG CTG CGC CGC CGG GAA AAT ACT GTA CGG TTA GA	0.78	[152]
Nucleolin (ALL)	AS1411	GGT GGT GGT GGT TGT GGT GGT GGT GG	-	[153]
CXCL12, MS-5 cells (CLL)	NOX-A12	GCG UGG UGU GAU CUA GAU GUA UUG GCU GAU CCU AGU CAG GUA CGC	0.2	[154,155]
HL60 cells (AML)	KH1C12	ATC CAG AGT GAC GCA GCA TGC CCT AGT TAC TAC TAC TCT TTT TAG CAA ACG CCC TCG CTT TGG ACA CGG TGG CTT AGT	4.5 ± 1.6	[156]
Sigles-5, NB4 cells (AML)	K19	AAG GGG TTG GGT GGG TTT ATA CAA ATT AAT TAA TAT TGT ATG GTA TAT TT	12.37	[157]
IgM, Ramos cells (BL)	TD05	ACC GGG AGG ATA GTT CGG TGG CTG TTC AGG GTC TCC TCC CGG TG	7.9–359	[158,159]
		**Breast cancer**		
Nucleolin MCF-7 cells	AS1411	GGT GGT GGT GGT TGT GGT GGT GGT GG	-	[153]
Epitope peptide HER2	HB5	AAC CGC CCA AAT CCC TAA GAG TCT GCA CTT GTC ATT TTG TAT ATG TAT TTG GTT TTT GGC TCT CAC AGA CAC ACT ACA CAC GCA CAT G	18.9	[160]
HER2 protein	HeA2_3	TCT AAA AGG ATT CTT CCC AAG GGG ATC CAA TTC AAA CAG C	6.2	[161]
HER2	H2	GGG CCG TCG AAC ACG AGC ATG GTG CGT GGA CCT AGG ATG ACC TGA GTA CTG TCC	270	[162]
SK-BR-3 cells	S6	TGG ATG GGG AGA TCC GTT GAG TAA GCG GGC GTG TCT CTC TGC CGC CTT GCT ATG GGG	94.6	[163]
EpCAM	SYL3C	CAC TAC AGA GGT TGC GTC TGT CCC ACG TTG TCA TGG GGG GTT GGC CTG	38 ± 9	[164]
		**Pancreatic cancer**		
Nucleolin	APTA-12	GGT GGT GGT GGT TZ*T GGT GGT GGT GG	14.37 ± 8.93	[165]
MMP14 proteinase MIA PaCa-2, PANC-1 cell lines	M17	AGG GCC CGA CGT GAC GGC ACG TCG GAT ATC TCA TGC GTG T	4.98 ± 1.26	[166]
		**Colorectal cancer**		
DHX9, RNA helicase	S-1	GCC CAG CAT GCA TTA CTG ATC GTG GTG TTT GCT TAG CCCA	140	[167]
VEGF	SL2B	TTT TTT TTT ACA TTC CTA AGT CTG AAA CAT TAC AGC TTG CTA CAC GAG AAG AGC CGC CAT AGTA	-	[168]
CEA CA50 CA72-4	CAA01 CA50 A02 CA72-4 A01	GGG UCG UGU CGG AUC CAG GCA CGA CGC AUA GCC UUG GGA GCG AGG AAA GCU UCU AAG GUA ACG AU GGG UCG UGU CGG AUC CAG CUC GAA AGU GGG CUG GCG AUG UGU CCC GAA GCU UCU AAG GUA ACG AU GGG UCG UGU CGG AUC CUG CGA AGG GGG GCA GAG GUU UGA CGC GAG AAA GCU UCU AAG GUA ACG AU	16.5 30.7 52.7	[169]
		**Lung cancer**		
Lung cancer marker	APT-43	CTA TAG CAA TGG TAC GGT ACT TCC TCT CAG GTG GGT GTA TGT GGG CTC CCT TTA CTG ATT GGG TCA AAA GTG CAC GCT ACT TTG CTAA	64.0 ± 3.6	[170]
A549 cell line	-	GGT TGC ATG CCG TGG GGA GGG GGG TGG GTT TTA TAG CGT ACT CAG	-	[171]
		**Ovarian cancer**		
CD44, SKOV3, IGROV, A2780 cell lines	TA6	TTG GGA CGG TGT TAA ACGA AAG GGG ACG AC	187.0 ± 30.6	[172]
CA125	CA125.1	AAA AUG CAU GGA GCG AAG GUG UGG GGG AUA CCA ACC GCG CCG UG	4.13	[173]
CD70	Apt928	GCT GTG TGA CTC CTG CAA GCG GGA AGA GGG CAG GGG AGG GAG GGT GAC GCG GAA GAG GCA AGC AGC TGT ATC TTG TCT CC	66	[174]
A2780, SKOV3 cells	R13	CTC TAG TTA TTG AGT TTT CTT TTA TGG GTG GGT GGG GGG TTT TT	50	[175]
A2780T cells	HF3-58 HA5-68	TTG GAG CAG CGT GGA GGA TAT GCT TTC CGA CCG TGT TCG TTT GTT ATA ACG CTG CTC C TTA AGG AGC AGC GTG GAG GAT ATC GGT GTT TAT GGT GTC TGT CTT CCT CCA GTT TCC TTC TGC GCC TT	0.30 ± 0.24 4.5 ± 1.6	[176]

ALL—acute lymphoblastic leukemia; CLL—Chronic lymphocytic leukemia; AML—Acute myeloid leukemia; BL—Burkitt’s lymphoma; VEGF—vascular endothelial growth factor. Z*—gemcitabine is a first-line chemotherapy agent for the treatment of pancreatic cancer.

**Table 3 polymers-13-00341-t003:** Antibodies used for specific drug delivery to various types of cancer.

Antibody	Drug	Application/Target	Linker	Ref.
		**Tumor vasculature**		
anti-CD276	pyrrolobenzo-diazepine	tumor vasculature in CD276^+^ tumors	maleimide linker	[182]
anti-TM4SF1	auristatin (LP2)	TM4SF1^+^ tumor vasculature	noncleavable maleimido-caproyl	[183]
anti-PTK7 (PF-06647020)	auristatin (Aur0101)	tumor initiating cells/tumor vasculature	protease-cleavable valine-citrulline	[184]
		**Breast cancer**		
anti- HER2 (Trastuzumab)	trastuzumab emtansine	Metastatic breast cancer	Protease-cleavable tetrapeptide linker	[185]
CR011	monomethylauristatin E	breast cancer/glycoprotein NMB	Dipeptide	[186]
anti-LIV-1 (Ladiratuzumab)	monomethyl auristatin E	Triple-negative breast cancer/LIV-1	Protease-cleavable dipeptide	[187,188]
anti-Trop-2 (Sacituzumab)	SN-38	triple-negative breast cancer/Trop-2	Hydrolysable link, with short polyethylene glycol moiety(CL2A)	[188,189]
		**Lymphoma**		
anti-CD33 (gemtuzumab ozogamicin)	calicheamicin	acute myeloid lymphoma	Acid-labile (N-acyl hydrazine)	[190]
anti-CD30 (brentuximab vedotin)	monomethyl auristatinE (MMAE)	relapsed Hodgkin’s lymphoma/ anaplastic large cell lymphoma	cathepsin cleavable	[190]
		**Neurological cancer**		
anti-EGFR (depatuxizumab mafodotin)	monomethyl auristatinF (MMAF)	glioblastoma	noncleavable maleimido-caproyl	[190]
anti-ALK (CDX-0125-TEI)	thienoindole (NMS-P945	human neuroblastoma xenograft in mice neuroblastomas	dipeptidic/cleavable	[191]
		**Other**		
anti–PD-L1 (MPDL3280A)	-	inhibit PD-L1 interactions with both PD-1 and B7-1 (f.e. metastatic bladder cancer)	-	[192]
Anti-AXL (enapotamab vedotin)	monomethyl auristatinE (MMAE)	non-small lung cancer, ovarian, cervical, endometrial, thyroid, melanoma	Protease-cleavable valine-citruline	[190,193]
anti-PSMA (MEDI3726)	pyrrolobenzo-diazepine	Metastatic prostate cancer	cathepsin cleavable	[190,194]

## Data Availability

Not applicable.

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
