# Peer review of "Polymer Nanoparticles and Nanomotors Modified by DNA/RNA Aptamers and Antibodies in Targeted Therapy of Cancer"

_polymers, 2021, doi:10.3390/polym13030341_

Round 1
Reviewer 1 Report
The manuscript entitled Polymer Nanoparticles and Nanomotors Modified by DNA/RNA Aptamers and Antibodies in Targeted Therapy of Cancer for Polymers Journal.
The concept of the manuscript is novel, fits and suitable to publish in Polymers Journal. This review addressed all details of Polymer Nanoparticles and Nanomotors Modified by DNA/RNA Aptamers and Antibodies for cancer treatment. The review is generally well written and clearly presented however still need to address some comments to improve the quality of manuscript and thus require minor revision before its acceptance.
- Provide a nice graphical abstract representing the overview of the MS with key highlights in my opinion for review article it should be mandatory. Give full form of abbreviation in the abstract as well as in whole manuscript.
- In the introduction section, write the novelty of the work and the problem statement clearly. Give details of usefulness of NPs in the cancer treatment is essential. Line no 69 to line no 77 give suitable reference. More discussion of polymeric nanocomposites and their usefulness is essential.
- Fig 3 give details of inside the figure, Figure 4 text size should be higher and Fig 5 should be modify not clearly visible.
- Up to section 3 the text information need to be shorten very well since this is not a major focus of the review article. Figure 7 should be in color form.
- Advantages, disadvantage and toxicity issues should be discussed.
- More details about the practical applications and future research perspectives and challenges is expected.
- The conclusion of the study is not discussed with the specific output obtained from the study, it could be modified with precise outcomes with a take home message.
Author Response
Comment: The manuscript entitled Polymer Nanoparticles and Nanomotors Modified by DNA/RNA Aptamers and Antibodies in Targeted Therapy of Cancer for Polymers Journal.
The concept of the manuscript is novel, fits and suitable to publish in Polymers Journal. This review addressed all details of Polymer Nanoparticles and Nanomotors Modified by DNA/RNA Aptamers and Antibodies for cancer treatment. The review is generally well written and clearly presented however still need to address some comments to improve the quality of manuscript and thus require minor revision before its acceptance.
Response: We are grateful to the reviewer for positive opinion as well as for very useful comments that allowed us to improve manuscript.
Comment: Provide a nice graphical abstract representing the overview of the MS with key highlights in my opinion for review article it should be mandatory. Give full form of abbreviation in the abstract as well as in whole manuscript.
Response: The graphical abstract and highlights were provided. The full abbreviations were included as recommend.
Comment: In the introduction section, write the novelty of the work and the problem statement clearly. Give details of usefulness of NPs in the cancer treatment is essential. Line no 69 to line no 77 give suitable reference. More discussion of polymeric nanocomposites and their usefulness is essential.
Response: The novelty of the work, corresponding reverences and other requested information has been included in the Introduction according to reviewer's comments.
Comment: Fig 3 give details of inside the figure, Figure 4 text size should be higher and Fig 5 should be modify not clearly visible.
Response: The figures were improved as suggested by reviewer.
Comment: Up to section 3 the text information need to be shorten very well since this is not a major focus of the review article. Figure 7 should be in color form.
Response: It was difficult to shorten the section 2 as the second reviewer recommended inclusion of additional information concerning polymer nanoparticles. Figure 7 was replaced by color one as requested by reviewer.
Comment: Advantages, disadvantage and toxicity issues should be discussed.
Response: The corresponding discussion has been included in the revised manuscript.
Comment: More details about the practical applications and future research perspectives and challenges is expected.
Response: Corresponding discussion including clinical applications and future perspectives were included in part 7.
Comment: The conclusion of the study is not discussed with the specific output obtained from the study, it could be modified with precise outcomes with a take home message.
Response: The Conclusion has been improved as recommended by reviewer.
Reviewer 2 Report
Hianik and coworkers summarized recent examples about polymeric nanoparticles and nanomotors with apamter/antibody conjuations. The review is well organized, covering from basic concepts to applications. A few issues shall be addressed.
- Nanomotor was discussed as a major portion throughout the manuscript. However, the abstract does not seem to properly emphasize this point. Please revise.
- Section 2 mentioned several methods for the preparation of polymeric nanoparticles. The only issue is that most of them did not start with a clear definition on what the method is and how to conduct the method. For example, Line 159, the introduction of nanoprecipitation was only about the year and the lead author. Details of the method are suggested to come right after this. Similar problem exists with most of the methods in this section. Please elaborate.
- References are missing in Section 2. For example, several recent references on nanoprecipitation are suggested to be referred: DOI: 10.1016/j.matlet.2019.127018; DOI: 10.1002/anie.201913539; DOI: 10.1021/acs.macromol.9b02595; DOI: 10.1021/acs.iecr.9b04747. Similarly, references for emulsions are missing: DOI: 10.2174/187221112802652606; DOI: 10.1002/mabi.201900063; DOI: 10.1088/1748-6041/5/6/065002; DOI: 10.1007/s003960050130.
- Table 2 demonstrates a nice example on summarizing the existing apatmers for several cancer cells. A similar table for the choice of antibodies should also be included.
Author Response
Hianik and coworkers summarized recent examples about polymeric nanoparticles and nanomotors with aptamer/antibody conjugations. The review is well organized, covering from basic concepts to applications. A few issues shall be addressed.
Response: We are very grateful to the reviewer for positive opinion and for very useful comments that allowed us to improve of the manuscript.
Comment: Nanomotor was discussed as a major portion throughout the manuscript. However, the abstract does not seem to properly emphasize this point. Please revise.
Response: The abstract was revised as recommended.
Comment: Section 2 mentioned several methods for the preparation of polymeric nanoparticles. The only issue is that most of them did not start with a clear definition on what the method is and how to conduct the method. For example, Line 159, the introduction of nanoprecipitation was only about the year and the lead author. Details of the method are suggested to come right after this. Similar problem exists with most of the methods in this section. Please elaborate.
Response: The manuscript has been improved according to reviewer's suggestions. More details were included on the preparation of nanoparticles.
Comment: References are missing in Section 2. For example, several recent references on nanoprecipitation are suggested to be referred: DOI: 10.1016/j.matlet.2019.127018; DOI: 10.1002/anie.201913539; DOI: 10.1021/acs.macromol.9b02595; DOI: 10.1021/acs.iecr.9b04747. Similarly, references for emulsions are missing: DOI: 10.2174/187221112802652606; DOI: 10.1002/mabi.201900063; DOI: 10.1088/1748-6041/5/6/065002; DOI: 10.1007/s003960050130.
Response: The references were included as recommended.
Comment: Table 2 demonstrates a nice example on summarizing the existing aptamers for several cancer cells. A similar table for the choice of antibodies should also be included.
Response: New Table 3 has been prepared that is related to application of antibodies in targeted drug delivery.